# Host Immunity Alters Community Ecology and Stability of the Microbiome in a *Caenorhabditis elegans* Model

Megan Taylor,[a] ORCID N. M. Vega[a]

[a]Biology Department, Emory University, Atlanta, Georgia, USA

**ABSTRACT** A growing body of data suggests that the microbiome of a species can vary considerably from individual to individual, but the reasons for this variation—and the consequences for the ecology of these communities—remain only partially explained. In mammals, the emerging picture is that the metabolic state and immune system status of the host affect the composition of the microbiome, but quantitative ecological microbiome studies are challenging to perform in higher organisms. Here, we show that these phenomena can be quantitatively analyzed in the tractable nematode host *Caenorhabditis elegans*. Mutants in innate immunity, in particular the DAF-2/insulin growth factor (IGF) pathway, are shown to contain a microbiome that differs from that of wild-type nematodes. We analyzed the underlying basis of these differences from the perspective of community ecology by comparing experimental observations to the predictions of a neutral sampling model and concluded that fundamental differences in microbiome ecology underlie the observed differences in microbiome composition. We tested this hypothesis by introducing a minor perturbation into the colonization conditions, allowing us to assess stability of communities in different host strains. Our results show that altering host immunity changes the importance of interspecies interactions within the microbiome, resulting in differences in community composition and stability that emerge from these differences in host-microbe ecology.

**IMPORTANCE** Here, we used a *Caenorhabditis elegans* microbiome model to demonstrate how genetic differences in innate immunity alter microbiome composition, diversity, and stability by changing the ecological processes that shape these communities. These results provide insight into the role of host genetics in controlling the ecology of the host-associated microbiota, resulting in differences in community composition, successional trajectories, and response to perturbation.

**KEYWORDS** *Caenorhabditis elegans*, immunocompromised hosts, microbial communities, microbial ecology

Host-associated microbiomes are increasingly recognized as ecological systems, where interactions among microbes and between microbes and their host are important for shaping community composition, structure, and function (1, 2). As this understanding has developed, a search for the ecological principles that define these communities has ensued (3, 4).

There is particular interest in understanding the sources and consequences of variation in host-associated microbiotas. Host-associated microbiomes associated with any given body site can vary considerably between individuals and within individuals over time (5, 6). Some of this variation is attributable to the stochastic processes of colonization and drift experienced by any open ecological system (7–11). However, there is a gathering consensus that host-associated microbial communities are shaped by deterministic processes, including filtering (selection

Address correspondence to N. M. Vega, nvega@emory.edu.

Innate immunity changes microbiome ecology and stability in a nematode model.

**TABLE 1** *C. elegans* lineages used in this study[a]

| Worm Strain | Mutation | Description |
|---|---|---|
| N2 | | Wild type |
| DA573 | *eat-14(ad573)* X | |
| DA597 | *phm-2(ad597)* I | |
| CF1038 | *daf-16(mu86)* I | DAF-2/IGF defective |
| CB1370 | *daf-2(e1370)* III | DAF-2/IGF up-regulated |
| CF1449 | *daf-16(mu86)* I; *daf-2(e1370)* III; muEx176 [*daf-16*p::GFP::*daf-16* + *rol-6*(su1006)] | Double mutant; pick non-GFP non-rollers for digests |
| NU3 | *dbl-1(nk3)* V | TGF-β defective |
| BW1940 | *ctIs40X* [*dbl-1*(+) + *sur-5*::GFP] | TGF-β over-expression |
| AU37 | *glp-4(bn2)* I; *sek-1(km4)* X | p38 MAPK defective |
| JT366 | *vhp-1(sa366)* II | p38 MAPK over-expression |
| SS104 | *glp-4(bn2)* I | AU37 control strain |

[a]Pharyngeal mutants are highlighted in blue, innate immune mutants are highlighted in yellow, and control strains are highlighted in gray.

of colonists by the host) and competitive and cooperative interactions among microbes (10, 12, 13).

Some of this variation can be traced to genetic differences between hosts (14–16). However, the effect of host genetic variation on the ecology of microbiome communities remains poorly understood. Much of the existing data focus on interactions between individual commensal bacteria and their specific hosts, although there is increasing interest in understanding the mechanisms by which a host can control the composition of its microbiome (17–19). If host genetics affect the ecological processes of community assembly, differences between hosts can result in differences in microbiome succession, composition, and stability properties due to differences in the underlying ecology of these communities. Understanding how the host environment shapes the ecological dynamics of microbiome communities will be important for determining how communities in different hosts might respond differently to normal perturbations (e.g., changes in nutrient availability, exposure to innocuous microbes, and host circadian rhythms) and pathological events (e.g., pathogen invasion, drug or toxin exposure, and host disease or trauma).

Here, we present experimental evidence that differences in host genetics can alter the ecological dynamics of microbiome communities, resulting in differences in assembly, succession, and response to perturbation. Using a minimal native microbiome of the nematode *Caenorhabditis elegans*, we colonized N2 wild-type hosts and well-characterized mutant strains under highly controlled conditions to determine the effects of host genetics on the structure and dynamics of intestinal communities.

## RESULTS

**Effects of host genetics on microbiome assembly.** In these experiments, germ-free, reproductively sterile adult *C. elegans* organisms from N2 (wild type) and selected mutant strains (Table 1) were colonized from an artificially constructed metacommunity of eight bacterial strains. These bacterial strains represent a taxonomically and functionally diverse subset of isolates from a wild *C. elegans* microbiome (20, 21) (see Materials and Methods). Each possessed a unique colony morphology when cocultured on agar plates, allowing CFU counts for each species to be taken from mixed communities (Fig. S1).

Communities in the N2 intestine showed distinct ecological patterns despite high variability (Fig. 1A; Data Set S1; Table 1). While total colonization ranged across nearly two orders of magnitude, from 1,680 to 67,200 CFU/worm, community composition did not change dramatically across this range (Fig. S2). Overall, these communities

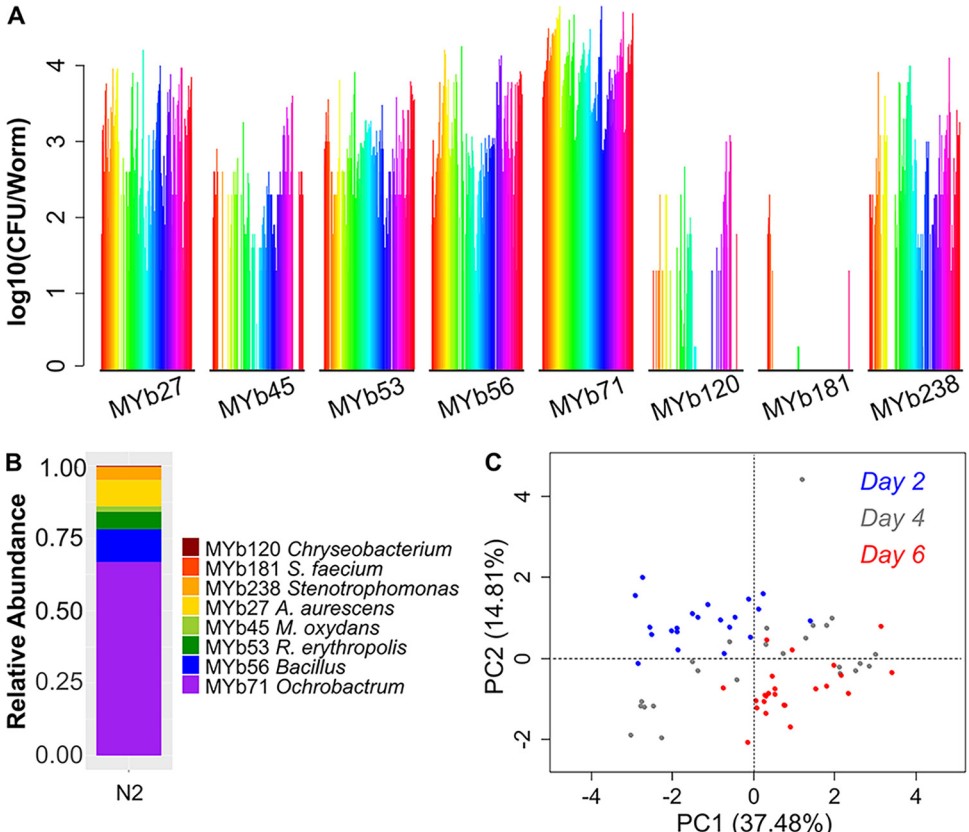

**FIG 1** Eight-species microbial communities in the N2 intestine show distinct trends and variation. N2 worms were sampled from 12 independent experiments conducted over the course of ~1 year. Each individual experiment contains data for 12 to 36 individual N2 worms taken from a single well. (A) CFU-per-worm data for the full data set of individual hosts ($n = 164$) after 4 days of colonization with this eight-species bacterial consortium. Each color represents a single host, and data are grouped by bacterial species to illustrate trends in abundance. (B) Average relative abundance of each bacterial species across all individual N2 worms. (C) Principal-component analysis (PCA) of community composition over time in N2 worms across a 6-day time series of colonization (24 individual worms/day, destructive sampling of individual hosts).

showed high frequencies of MYb71 (*Ochrobactrum*) (Fig. 1B), consistent with prior observations that this genus tends to dominate communities in lab-colonized worms (20). Community composition differed in worms sampled at different time points in colonization (Fig. 1C), and day 4 of colonization appears to represent midsuccession in these experiments. We therefore opted to measure communities after 4 days, which we hypothesized would provide useful data on differences in ecological dynamics across host strains, albeit while failing to maximize compositional differences between lineages.

Next, we explored the effects of host mutations on the composition of the microbiome (Fig. 2; Data Set S1; Table 1; Fig. S3). All worm mutants were colonized under the same conditions used for N2 (above), and preliminary testing indicated differences between host strains (analysis of similarity [ANOSIM] based on Bray-Curtis differences, 9,999 permutations, $R = 0.27$, $P < 0.0001$; permutational multivariate analysis of variance [PERMANOVA], 999 permutations, $F = 54.624$, $P = 0.001$). First, we analyzed grinder-defective mutants (*eat* and also *phm-2* mutants), which have known alterations in their interactions with bacteria (22, 23). Bacterial communities in the mild grinder mutant *eat-14* were very similar to those in N2. While communities in the severe grinder *phm-2* mutant were large compared to those in N2 (median CFU/worm, 36,190 versus 13,000), composition was within the range observed for N2 (Fig. 2A). Increased permissiveness of the defective grinder did not substantially affect community assembly, consistent with previous results (24).

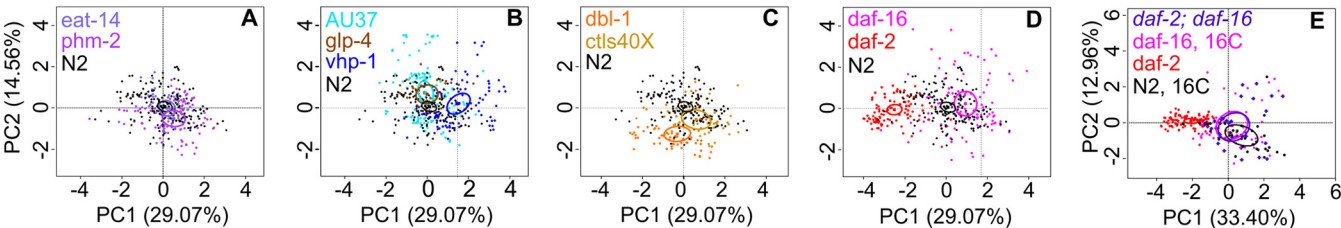

**FIG 2** PCA of intestinal community composition in N2 wild-type ($n = 164$; black points) and mutant hosts colonized from a uniform metacommunity of eight bacterial species from the *C. elegans* native microbiome. (A to D) Subplots of the large ordination shown in its entirety in Fig. S3B; all worms except the constitutive *daf-2* dauer mutant were grown to adulthood at 25°C. (A) Mild (*eat-14*; $n = 69$) and severe (*phm-2*; $n = 78$) grinder mutants; (B) p38 MAPK pathway-defective (AU37; $n = 115$) and derepressed (*vhp-1*; $n = 84$) mutants, with a *glp-4* ($n = 36$) control for the AU37 strain; (C) TGF-$\beta$ defective mutant (*dbl-1*; $n = 69$) and overexpression construct (*ctls40*; $n = 48$); (D) DAF-2/IGF defective (*daf-16*; $n = 100$) and derepressed (*daf-2*; $n = 98$) mutants. (E) Separate ordination, based on data from worms grown to adulthood at 16°C. When all worm hosts are grown to adulthood at 16°C, the double mutant CF1449 is indistinguishable from the *daf-16* single mutant. Growth to adulthood at 16°C alters the later acquisition of microbial communities when adult worms (N2 and *daf-16* strains) are colonized at 25°C. All data represent the results of single-worm digests and plating after 4 days total colonization on the uniformly distributed synthetic eight-species bacterial consortium. Ellipses for center of mass of the data associated with each host strain were generated using *coord.ellipse* (R package *FactoMineR*) with a confidence of 0.9999.

We then explored the effect of the well-conserved innate immune system of *C. elegans* (25, 26) on composition of the microbiome. All three pathways of innate immunity (p38, transforming growth factor $\beta$ [TGF-$\beta$], and DAF-2/insulin growth factor [IGF]) have been shown to be differentially expressed in worms raised on a complex microbiota compared with *Escherichia coli* (24). We found that bacterial communities in innate immunity mutants showed apparent differences from wild-type N2 (Fig. 2B to D), but the relationship between community composition and immune function in this host is complex. Neither gain nor loss of function in innate immune pathways had a consistent effect on microbiome composition. As previously described (24), loss of function in p38 mitogen-activated protein kinase (MAPK) (AU37) produced microbial communities that compositionally resembled those in a *glp-4*(*bn2ts*) control and in N2; however, derepression of this pathway (*vhp-1*) resulted in a microbiome that appeared to diverge from that of N2 (Fig. 2B). Conversely, loss of function in TGF-$\beta$ (*dbl-1*) produced communities that diverged from those of the wild type, as previously observed (24), while overexpression (*ctls40*) had no apparent effect (Fig. 2C). Finally, both loss of function (*daf-16*) and derepression (*daf-2*) of the DAF-2/IGF pathway produced marked effects on microbiome composition and variation (Fig. 2D).

We confirmed that microbial colonization under these conditions was associated with differential activation of *daf-16* using a fluorescent reporter assay (Fig. S4). Intestinal community composition of the double mutant (*daf-2*; *daf-16*) was indistinguishable from that of the *daf-16* single mutant (Fig. 2E), suggesting that the effects of the *daf-2* mutation on microbiome composition are mediated through dysregulation of DAF-16. Further, when N2 and *daf-16* worms are grown to adulthood at 16°C instead of 25°C to allow a more direct comparison with the double mutant, the communities acquired by these adults differ from those in worms of the same genotype grown to adulthood at 25°C, when colonized at 25°C according to the standard protocol (Fig. S3C and D).

Next, we sought to establish ecological mechanisms underlying the observed differences in microbiome composition. As previously indicated, we chose to sample communities after 4 days of colonization to capture midsuccessional ecology. N2 wild-type worms showed a negative trend in the relationship between diversity (measured as Shannon H) and total microbiome size (CFU per worm in individual worms) (Fig. 3A), although the considerable compositional and structural variation in these communities (Fig. 1) results in a poor overall fit for the simple linear model (Table S1). Pharyngeal mutants showed diversity-size relationships similar to those of N2 (Fig. 3B), consistent with the compositional similarities observed (Fig. 2). Among the innate immune system mutants (Fig. 3C to E), most of the lineages that showed apparent compositional differences from N2 also showed differences in the diversity-size relationship (27) (*vhp-1* mutant with likelihood ratio test $P = 1.94 \times 10^{-4}$; *daf-2* mutant $P = 5.55 \times 10^{-7}$; and *daf-16*

mSystems®

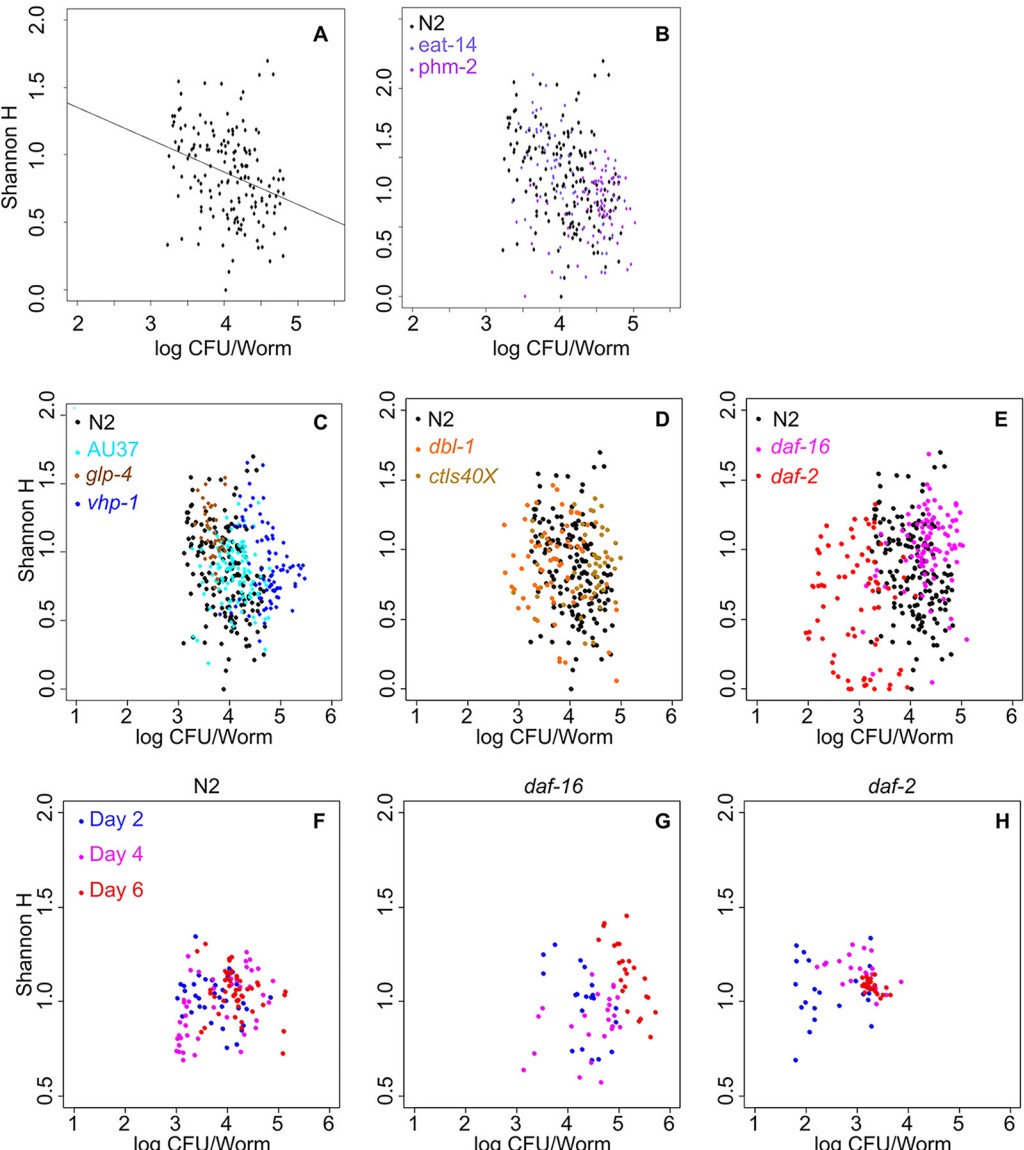

**FIG 3** Successional ecology is altered by host genetic background. (A to E) Shannon diversity versus $\log_{10}$(CFU/worm) for the data set shown in Fig. 2 and Fig. S3. (A) N2 wild-type worms ($n = 164$) show a generally negative diversity-population size trend [linear fit $y = 1.83 - 0.24x$; $R^2$(adj) = 0.09; $P = 8.748e-5$]. (B) Mechanical mutants (*eat-14* and *phm-2* mutants) show a trend similar to that of the wild type. (C to E) The diversity-population size relationship in intestinal bacterial communities differs across host lineages. Mutants in the (C) p38 MAPK, (D) TGF-*β*, and (E) DAF-2/ IGF pathways show trends that diverge in some cases (see Table S1) from that observed in N2 (black dots). Note that Shannon diversity in these experiments has a maximum at ln(8) = 2.08. (F to H) Time series of the diversity-population size relationship indicate that succession is altered in DAF-2/IGF mutant hosts. Intestinal communities were quantified for individual worms ($n = 24$ worms per time point per condition) of (F) N2, (G) *daf-16*, and (H) *daf-2* lineages after 2, 4, and 6 days of colonization on the uniform eight-species metacommunity. CFU-per-worm data underlying panels F to H are shown in Fig. S5.

mutant $P = 1.61 \times 10^{-9}$) (Table S1), suggesting that differences in competition between gut bacteria and the process of ecological succession underlie the observed differences in community composition. (Note that *dbl-1* adults are physically smaller than N2 adults, which likely explains the smaller populations observed in these hosts [28]. Also, note that several of these linear fits indicate no significant relationship between Shannon diversity and total microbiome size, and the large *β*-diversity within host lineages is expected to be a factor here.)

If ecological succession differs between *C. elegans* mutant strains, there should be differences in microbiome development over time. Based on observed divergences

from the wild type, we chose to use DAF-2/IGF mutant hosts in these experiments. Here, we colonized adult worms from the N2, *daf-16*, and *daf-2* lineages with the uniform eight-species metacommunity and quantified intestinal communities at 48-h intervals during community development. In N2 hosts, we observe convergence of communities over time, as previously described (Fig. 1C); by day 6 of community development, N2-associated intestinal communities had largely converged to a fairly wide but well-defined range ($10^{3.5–4.5}$ CFU/worm; $0.5 < H < 1.5$), and the negative diversity-population size relationship had diminished, suggesting a later stage of succession (Fig. 3F; Fig. S5A to C). *daf-16* hosts displayed large populations which continued to increase in size and diversity over the observed period (Fig. 3G; Fig. S5D to F), while *daf-2* hosts showed convergence to smaller microbiomes consisting mainly of three dominant bacteria (Fig. 3H; Fig. S5G to I). These data suggest that differences in ecological succession underlie the observed differences in community composition. However, it is not clear what mechanism(s) underlies these differences.

**DAF-2 signaling alters host control of microbiome assembly.** Genetic differences between hosts could alter microbiome ecology by altering the efficiency of host selection during colonization (environmental filtering), by changing the interspecies interactions among bacteria, or both. This should be associated with differences in interspecies correlations within these communities. In this scenario, the *daf-2* mutant is hypothesized to represent a highly selective environment; due to strong interactions with the host, bacterial density rarely rises to the level where competition between strains is a major influence. The anticipated outcome would be strong compositional convergence enforced by environmental filtering and weak negative correlations among bacterial species. Conversely, the *daf-16* mutant is hypothesized to represent an environment with poor active selection by the host. A lack of host control could produce the large and diverse communities observed and should result in stronger-than-expected correlations (negative and/or positive) among bacterial strains due to increased interaction in the intestine.

To test this hypothesis, we first calculated the Spearman correlations for pairs of bacterial species within each of N2, *daf-16*, and *daf-2* hosts, using the large data set underlying Fig. 2. Intestinal communities in all host lineages showed a range of correlations, and the distributions of interspecies correlations differed across hosts. The *daf-16* communities showed a broad distribution of correlations relative to N2, with fewer near-zero interactions, while *daf-2* communities showed stronger positive correlations but weaker negative correlations than those in N2 (Fig. 4A to C; Fig. S6A to F), consistent with the hypothesis. Note that positive correlations in these data need not reflect true positive interactions between species; a positive relationship between counts of two bacteria can simply reflect common species being common together, when the number of total bacteria varies among samples as it does in these data.

We therefore sought to quantify expectations for these distributions that take into account differences in sample size between lineages and in community size between individuals. As a full stochastic model of colonization would require estimation (empirical or otherwise) of a fairly large number of parameters, we opted for a simpler data-driven approach where we fitted each data set to a neutral sampling distribution.

The Dirichlet-multinomial distribution is a multivariate sampling distribution that has frequently been used to describe community data of this kind (29–31). In the neutral sampling process described by this model, bacterial species are allowed to have different probabilities of being "drawn" from the pool of potential colonists (in this scenario, where all species are present at equal levels in the metacommunity, this corresponds to different effective migration rates into the host) but species do not differ in their ability to fill space (compete) within the host. By comparing the empirical distributions of correlations to those produced under the neutral sampling assumption, we can determine whether our hypothesis—that the host immune system modifies the effective strength of interactions between bacteria in the gut, possibly by changing the relative importance of interactions with the host—is supported.

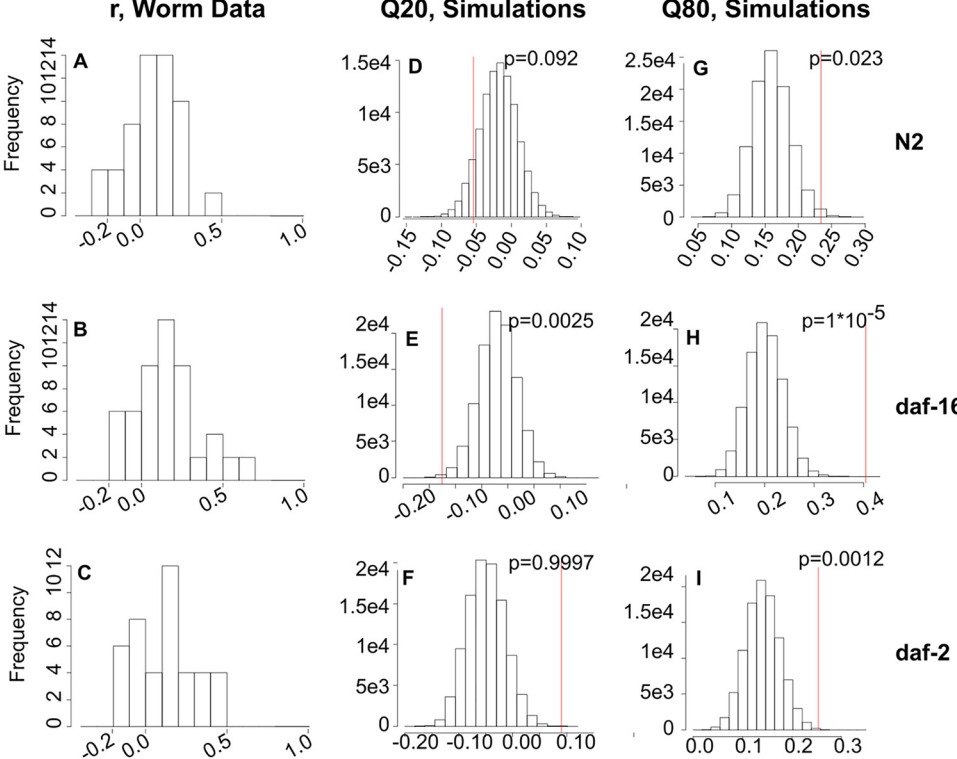

**FIG 4** Spearman correlations between bacteria in host-associated intestinal communities differ from the predictions of a neutral sampling model. (A to C) Histograms of Spearman correlation coefficients calculated from data for (A) N2 ($n = 164$), (B) *daf-16* ($n = 100$), and (C) *daf-2* ($n = 98$) intestinal communities, with a bin size of 0.1. (D to I) Histograms of Spearman coefficient 20th and 80th quantiles from data simulated using host lineage-specific parameterizations of the Dirichlet-multinomial model ($n = 10,000$ simulated data sets per condition). Red lines indicate the corresponding quantile from the empirical data, and *P* values indicate the one-tailed percentage of simulated data sets with a lower 20th quantile (D to F) or higher 80th quantile (G to I) than the empirical data.

We fitted this model to our data and used these parameterized distributions to generate simulated communities that replicate the structure of the real data, with the same number of hosts and the same number of bacteria in each host (Fig. S6 and S7). By generating a large number of simulated data sets ($n = 10,000$) for each host lineage, we can compare the correlations observed between bacterial species in real communities to those obtained from similar-quality simulated data where there are no true interactions between bacterial species. This allows us to assess the weight of evidence that interspecies interactions are important in each set of communities. Note that in this neutral model, real positive correlations are due simply to common species being common together, and negative interactions are spurious and appear solely due to limitations in sampling (Fig. S6G to J).

As expected, intestinal communities in the host diverge from the predictions of the neutral sampling model. In all three *C. elegans* strains, we observe more positive correlations than expected (Fig. 4G to I), suggesting that positive interactions among bacteria occurred within the worm host. Consistent with our hypothesis, the significance of this trend is strongest in the *daf-16* strain, where we hypothesized that host control is weakest and interactions among microbes strongest in shaping community composition. The N2 and *daf-16* hosts both show more negative correlations than expected (Fig. 4D and E); the trend is significant in *daf-16* communities but not in those of N2, again consistent with the predictions of our hypothesis. Interestingly, *daf-2* communities show significantly fewer negative correlations than expected (Fig. 4F), consistent with the prediction that negative interactions between bacterial species, such as competition, are less important in this apparently stringent host environment.

**DAF-2 signaling alters sensitivity of the microbiome to changes in colonization.** If the innate immune system actively shapes the microbiome by changing its ecological dynamics, intestinal communities in immune-mutant host strains should show different sensitivities to perturbation as a result. Specifically, we predict that communities in these hosts should react differently to a small change in the relative abundance of species in the metacommunity during community assembly, wherein one bacterial species is "dropped down" to 10% relative abundance. Here, we used a seven-species bacterial metacommunity; MYb56 (*Bacillus*) was removed due to high variation (24). In these experiments, the "rare" species will experience a drop in expected rate of migration into the host (7), resulting in reduced propagule pressure (number of individuals of this species introduced per event, or per unit of time) and thus increased variation in time to first successful colonization. We chose this setup (as opposed to a larger perturbation such as successive introduction of species) precisely because the mechanism we propose should be sensitive even to a small change in propagule pressure and because the increased variation in colonization by the rare species could increase community variation between individual hosts if priority effects are important (32–34).

We expect this perturbation to have specific effects if there are substantial priority effects (where the early composition of the community affects the species-specific probability of success of later colonists) mediated by interspecies interactions. We should see very little effect in a highly stringent host (*daf-2*) where environmental filtering dominates, as we expect bacterial species to colonize based on ability to survive this environment rather than on ability to interact with other colonizers. Conversely, this perturbation may have a large effect in an uncontrolled host (*daf-16*) where interactions between bacteria, rather than interactions with the host, are the (theorized) dominant force driving community assembly.

Other outcomes are possible. If priority effects are mediated through changes in the host state rather than through interactions between bacterial species, this perturbation should produce smaller changes in environments where host control is of less relative importance (N2 and *daf-16*) than in environments where the host is the dominant factor (*daf-2*). Alternately, if these systems are not driven by priority effects, the small perturbation introduced in these experiments should not alter community composition.

Our results (Fig. 5; Data Set S1) indicate that host genetics do affect the stability properties of the intestinal community. For N2 and *daf-16* intestinal communities, it is immediately clear that (with the possible exception of the drop-53 condition) (Fig. 5B and G) the perturbation has little or no effect on community composition. These host lineages show a broad but defined range of community compositions when colonized by the full experimental metacommunity (All-7), and the range is essentially recapitulated across conditions. This indicates that these communities are, as suggested previously, converging to a defined range and that this convergence is stable against the small perturbation applied here.

In contrast, communities in the *daf-2* intestine (Fig. 5K to O) occupied a relatively confined ordination space in the All-7 colonization condition, and several of the drop-down colonization conditions (notably, drop-71, -120, and -238) resulted in partial or total separation from the All-7 communities. Interestingly, run-to-run variation was considerable in *daf-2* mutant communities but not in N2 communities (Fig. S8). Even within the All-7 condition, it is clear that individual replicates for *daf-2* mutant communities represent different subsets of the total outcome space, while N2-associated communities tend to cover similar ranges across days. In two conditions (drop-71 and drop-120), one of three replicates diverged entirely from All-7, and in one condition (drop-238), all three replicates diverged from this baseline. These results are inconsistent with simple sources of experimental error (which would most likely have produced a single divergent run) and indicate that bacterial communities in the *daf-2* mutant have stability properties different from those of communities in N2 or *daf-16* hosts. Specifically, *daf-2* host communities appear to be both more deterministic

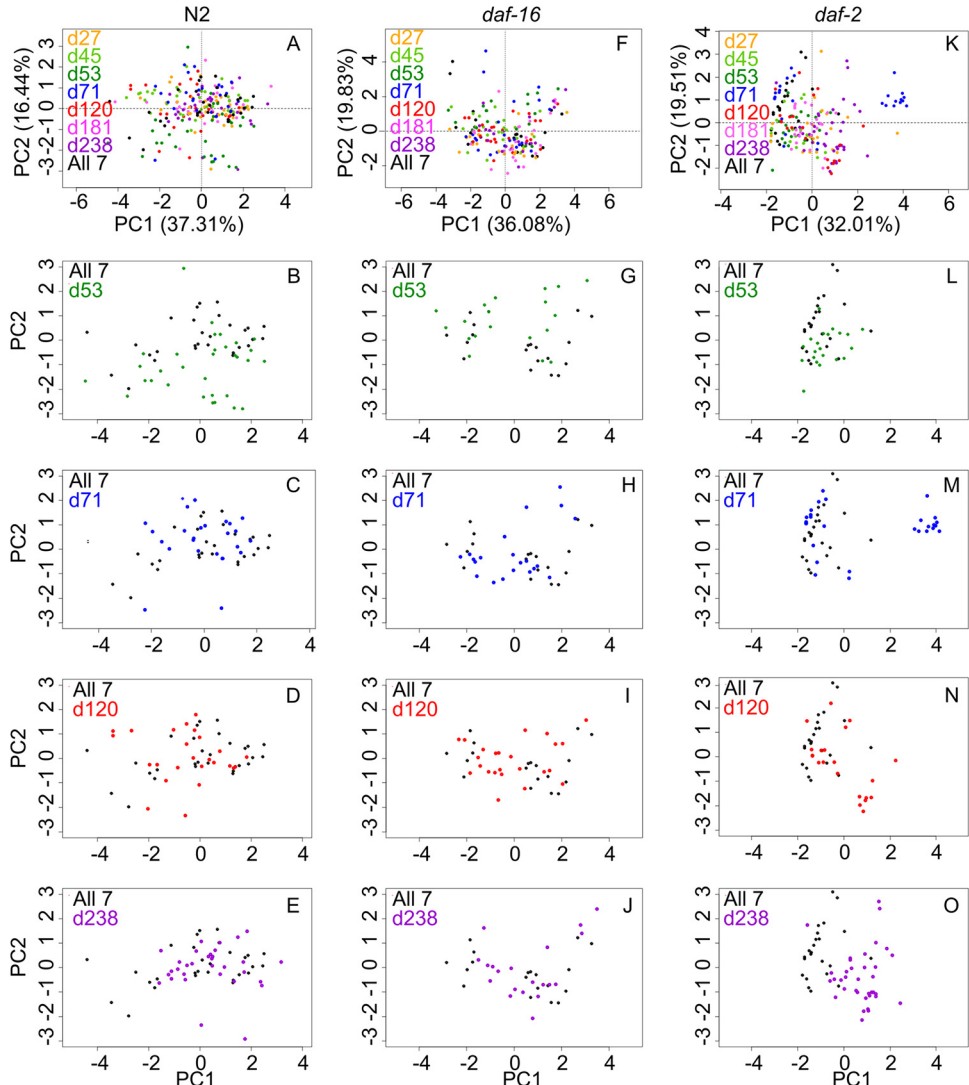

**FIG 5** Intestinal bacterial communities in different host lineages react differently to a shared perturbation. In these experiments, adult worms were colonized with an even metacommunity of seven bacterial species (MYb27, -45, -53, -71, -120, -181, -238 [All-7]) or a metacommunity where each species in turn is dropped (indicated with the letter "d") to 10% relative abundance. Worms were sampled at day 6 of colonization to allow time for communities to pass through early ecological succession. Data represent two (*daf-16* mutant) or three (N2 and *daf-2* mutant) independent runs, with 12 individual worms per host lineage/metacommunity combination; individual worms with <100 CFU (*daf-2*) or <1,000 CFU (N2 and *daf-16* mutant) were removed from data to minimize errors due to low colony counts.

(converging to a narrow range of outcomes for a given metacommunity) and more sensitive to initial conditions (such as inevitable run-to-run variation in the metacommunity [see the supplemental figure at 10.6084/m9.figshare.13317275]) than either N2 or the *daf-16* mutant. While these experiments are inadequate to fully describe the stability properties of these intestinal communities, these results are consistent with community landscapes for N2 and *daf-16* strains characterized by wide, stable attractors, while *daf-2* host communities appear to occupy a landscape with multiple alternate states (but see Discussion). Further, these results are collectively consistent with the hypothesis that assembly of N2 and *daf-16* mutant communities relies on interspecies interactions among bacteria, while assembly of communities in the *daf-2* host is controlled largely by the host, and host state changes (plausibly due in part to interactions with bacteria early in colonization) constrain the outcomes of community assembly.

## DISCUSSION

Here, we used a tractable model system to demonstrate how host genetics can alter microbiome composition and how the mechanisms underlying these compositional differences can result in differences in response to perturbation. We observe that changes in *C. elegans* immunity and stress response, in particular in the DAF-2/IGF pathway, are associated with changes in community assembly over time and in stability against perturbations in colonization conditions. These differences in stability result in different possible community states for a given starting condition, indicating that genetic differences in the host can affect the normal operating range and accessibility of alternate states for a microbiome.

These results demonstrate the importance of microbiome ecology for understanding dysbiosis. We observed that host-mediated differences in the ecological drivers of community dynamics result in differences in the availability and accessibility of alternate compositional states. These results suggest that differences in ecological dynamics can produce qualitative differences in propensity toward dysbiosis and the likelihood of return to "normal." By understanding the drivers of microbiome ecology, it may be possible to gain information about vulnerability to dysbiosis and to predict what types of intervention might be most effective at altering a given community.

It is important to note that these microbiome communities are likely not at a deterministic steady state. First, these are fundamentally not equilibrium systems; these communities are subject to continual disturbance from forces such as migration and physiological shifts in the host (35). Second, even if these communities were heading for a deterministic equilibrium, the relatively short life span of the host probably means that intestinal communities are in a transient state (7, 34, 36). Finally, the properties of the host and the intestinal environment are expected to change with age (37–39). Broadly, our results are consistent with previous descriptions of homeorhesis in synthetic communities (40), where a nonstationary system such as a microbiome converges to a trajectory rather than a deterministic point equilibrium, but it is also likely that our communities were in transient states. It is important to consider the underlying dynamics to understand how these results can be generalized, but the conclusions from the data are unaffected; we found distinct and replicable differences in community states and trajectories between different host strains, indicating differences in microbiome ecology driven by host genetics.

Our results are consistent with prior data on host genetics and intestinal community composition in *C. elegans*, but they differ in some particulars (24). Grinder-deficient worms (*tnt-3* mutants in the previous study, *phm-2* mutants here) show increased population sizes in the gut but no significant changes in community composition compared with N2. Likewise, both studies indicate that all three pathways of innate immunity may be invoked during response of this host to bacteria from its natural habitats. Berg et al. (24) observed that communities in *dbl-1* mutants (TGF-*β* defective) diverged strongly from those in N2, while we saw a smaller effect in this mutant; the previous study focused on differences in *Enterobacteriaceae* abundance, and this clade is not represented in our minimal eight-species community. However, we observed a measurable effect of *dbl-1* even in the absence of *Enterobacteriaceae*, and the previous study did not observe significant community effects in DAF-2/IGF mutants. It is plausible that the different bacterial community and/or the highly controlled liquid culture environment used here (compared with the experimental soil microcosm approach in the previous study) allowed us to detect these genetic effects.

The use of a highly controlled environment carries benefits and drawbacks. We chose to use a minimal eight-species community for tractability and to select our community to represent a taxonomically and functionally diverse subset of the native host microbiome to allow a range of possible interactions between species (20, 21). However, the small size of this community means that extrapolation to larger microbiota should be done with caution. We selected a well-controlled, well-mixed environment where worms were grown in liquid medium because it can be replicated

accurately, thus allowing us to minimize environmental variation and thereby increase our ability to detect genetic effects. However, as is always the case when a system is abstracted to increase control, this comes at the cost of reducing realism; among other factors, *C. elegans* is not adapted to swimming, and real environments are patchy. It remains unclear how environmental effects might combine with genetic factors to affect the stability properties of host-associated microbial communities. Although there is considerable prior work on the relative contribution of environmental and genetic factors on observed variation in microbiome composition (6, 41–44), at present there is neither theory nor experimental data sufficient for a systems-level explanation of how environment and genetics might act together, or how these interactions can be expected to affect response of microbiomes to perturbation.

Interpretation of these results is complicated by the pleotropic nature of worm genetics. For example, DBL-1 is important in development as well as immunity (45). Disruption of *glp-4* is known to have broad effects, including changes in protein translation (46); the *glp-4* mutant was used here as a control for AU37, and the *vhp-1* (*sa66*) mutant used here for upregulation of p38 does not have the *glp-4* mutation. Further, this *vhp-1* allele is probably hypomorphic (null alleles are lethal [47, 48]) with respect to p38 regulation. While our results suggest a complex role for innate immunity in regulation of the worm gut microbiome, further investigation of specific pathways is needed to disentangle immunity from other functions.

There are multiple mechanisms by which DAF-2/IGF signaling might control microbiome composition. This pathway has been implicated in a broad range of stress responses (49, 50), including oxidative stress and xenobiotics, in addition to response to bacterial pathogens, with *daf-2* mutants showing broadly increased resistance (51, 52). Total bacterial colonization can be affected, with *daf-2* mutants showing lower colonization than N2, but increased colonization is not always observed in *daf-16* mutants (53). Some data indicate that DAF-16 is not primarily involved in pathogen response but instead maintains a basal level of innate immunity, which regulates response to, and survival in the presence of, nonpathogenic or minimally pathogenic bacteria such as OP50 (54, 55). However, DAF-2/IGF is an insulin signaling pathway involved in satiety and quiescence behaviors, and we cannot presently rule out the possibility of behavioral differences, such as feeding rate, that might alter bacterial community assembly (56). Further, *E. coli* organisms colonizing DAF-2/IGF mutants show differences in gene expression compared with those colonizing wild-type N2 (57), suggesting that different bacterial factors are required for success; in our multispecies communities, this might result in differences in filtering based on metabolic capacity and/or differences in competitive ability based on metabolic shifts during colonization. Further, we observed that intestinal acidity is important in filtering and bacterial competition in *C. elegans* (58), and DAF-16 has been shown to promote acidification of intestinal lysosomes (59); if *daf-2* and *daf-16* mutants are at opposite ends of a spectrum of intestinal acidification, that might explain differences in the stringency of these host environments. Further research, including gene expression analysis in colonized hosts, is necessary to resolve these questions.

## MATERIALS AND METHODS

**Strains and culture conditions.** *Arthrobacter aurescens* (MYb27), *Microbacterium oxydans* (MYb45), *Rhodococcus erythropolis* PR4 (MYb53), *Bacillus* sp. strain SG20 (MYb56), *Ochrobactrum* sp. strain R-26465 (*anthropi*) (MYb71), *Chryseobacterium* sp. strain CHNTR56 (MYb120), *Sphingobacterium faecium* (MYb181), and *Stenotrophomonas* sp. (MYb238) were obtained from the Schulenburg lab (20). Bacterial strains were grown for 48 h at 25°C with shaking at 300 rpm in individual culture tubes with 1 ml of LB.

One hundred milliliters of 100% *C. elegans* axenic medium (AXN) was prepared according to published protocols (60) by autoclaving 3 g yeast extract and 3 g soy peptone (Bacto) in 90 ml water and subsequently adding 1 g dextrose, 200 $\mu$l of 5 mg/ml cholesterol in ethanol, and 10 ml of 0.5% (wt/vol) hemoglobin in 1 mM NaOH. To construct cultures to feed *C. elegans*, appropriate volumes of each strain to attain ~$10^8$ CFU/ml were washed in S-medium containing 1% AXN, and strains were centrifuged 2 min at 10,000 rpm to pellet and then resuspended in 1 ml S-medium plus 1% AXN.

Laboratory wild-type (N2) *C. elegans* and mutant strains (Table 1) were obtained from the *Caenorhabditis* Genetic Center, which is funded by NIH Office of Research Infrastructure Programs (P40

mSystems

OD010440). Nematodes were grown, maintained, and manipulated using standard techniques (61). Briefly, breeding stocks were maintained on nematode growth medium (NGM) plates with OP50 at 25°C (16°C for temperature-sensitive strains CB1370 [*daf-2*], AU37 [*glp-4*; *sek-1*], and SS104 [*glp-4*]) and synchronized using a standard bleach/NaOH protocol where eggs were allowed to hatch in M9 worm buffer overnight (~16 h) with shaking (200 rpm) at 25°C. Starved L1 larvae were transferred to 10-cm NGM plates containing lawns of *E. coli pos-1* RNA interference (RNAi) and incubated at 25°C for 3 days (most) or 16°C (the *daf-2* strain and CF1449; also N2 and the *daf-16* strain when specifically stated) to produce reproductively sterile adults; temperature-sensitive sterile strains AU37 and *glp-4* were grown to adulthood on OP50 under the same conditions. As some of the mutants used here develop to adulthood at different rates, care was taken to arrange and/or stagger synchronizations such that all strains within a given experiment reached adulthood at the same time. Worms were then transferred to liquid S-medium containing 200 $\mu$g/ml gentamicin, 50 $\mu$g/ml chloramphenicol, and 2× heat-killed OP50 (to trigger feeding) for 24 h, resulting in largely germfree adults (see the supplemental figure at 10.6084/m9.figshare.13317275). Adult worms were washed via sucrose flotation before colonization (61).

**Colonization of worms in liquid culture.** Bacterial strains were grown separately in 1 ml LB cultures for 48 h at 25°C and diluted to a uniform cell density of $10^8$ CFU/ml. Colonization was performed in well-mixed liquid media according to standard protocols (7) to ensure that all individuals experienced a uniform environment and had equal access to all potential colonists for the duration of colonization. Germfree adult worms were resuspended in S-medium plus 1% AXN to a concentration of ~1,000 worms/ml. Aliquots of 40 $\mu$l were pipetted into 96-well deep culture plates (1.2-ml well volume; VWR). A 20-$\mu$l portion of each bacterial suspension was added to each well (final volume, 200 $\mu$l). Plates were covered with Breathe-Easy sealing membranes and incubated with shaking at 200 rpm at 25°C.

After 2 days, worms were washed to remove external bacteria and refed on a freshly assembled metacommunity; this was done to enforce an evenly distributed community, maximize viability of potential colonists, and minimize bacterial evolution that might lead to unpredictable divergence across replicates. In this step, worms were washed twice in 1 ml of M9 worm buffer plus 0.1% (vol/vol) Triton X-100 (M9TX1), rinsed once with S-medium plus 1% AXN to remove surfactant, and resuspended in 100 $\mu$l of S-medium plus 1% AXN. Worms were fed as previously described in a fresh 96-deep-well plate.

**Mechanical disruption of colonized worms.** Prior to disruption, colonized worms were washed twice in M9TX1 and moved to an inert food source (heat-killed OP50) for 30 min to purge nonadhered bacteria from the gut and then washed and surfaced bleached to remove external bacteria before mechanical disruption of individual worms to retrieve intestinal communities. To clean the exterior cuticle, worms were rinsed twice with M9TX1, cooled for 15 min at 4°C to stop peristalsis, and bleached for 15 min at 4°C with 100 $\mu$l M9 plus 0.2% (vol/vol) commercial bleach (see the supplemental figure at 10.6084/m9.figshare.13317275). Worms were then rinsed twice with M9TX1 to remove bleach, treated with 100 $\mu$l of SDS/dithiothreitol (DTT) solution (965 $\mu$l M9 plus 5 $\mu$l SDS plus 30 $\mu$l 1 M DTT) for 20 min to permeabilize the worms, and washed once in 1 ml M9. A deep-well plate (2 ml square-well plate; Axygen) was prepared by adding ~0.2 g of sterile 36-mesh silicon carbide grit (Kramer Industries) and 180 $\mu$l of M9TX1 to each well. Worm samples were transferred to a 35-mm petri dish with 3 ml of M9TX1, and individual worms were pipetted manually into wells in 20 $\mu$l aliquots. The plate was covered with Parafilm and kept at 4°C for 1 h to reduce heat damage to bacteria. Parafilm-covered plates were capped with square silicon sealing mats (AxyMat) and disrupted by shaking in a Qiagen TissueLyser II at 30 Hz for 3 min. Plates were then centrifuged at 2,500 × *g* for 2 min to collect all material, resuspended by pipetting, and transferred to 96-well plates for 10-fold serial dilution in PBS (200-$\mu$l final volume per well).

**Measurement of bacterial communities.** Worm digests were dilution plated onto solid agar for quantification of intestinal communities; bacteria of different types were distinguished on the basis of colony morphology (Fig. S1). Serial dilutions of $10^{-1}$ and $10^{-2}$ of the digests, with the exception of strains DA597 and CB1370, were plated onto modified salt-free nutrient agar (3 g yeast extract, 5 g peptone, and 15 g of agar [Bacto] per liter). For strain DA597 (*phm-2*), dilutions of $10^{-2}$ and $10^{-3}$ were plated. Dilutions of $10^0$ and $10^{-1}$ were plated for CB1370 (*daf-2*). In all cases, 100-$\mu$l aliquots (of a 200-$\mu$l total volume) were plated onto individual 10-cm plates using bead shaking. Samples were incubated at 25°C for 48 to 72 h to allow distinct morphologies to develop. Colonies were then counted, and the number of CFU per worm was calculated for each sample. All worm strains in the data set are represented by multiple biological replicates, conducted on separate days, with 12 to 36 worms taken per host strain per experiment depending on the total number of strains in that experiment (Data Set S1; Table 1).

**Expression of *daf-16*.** *C. elegans* carrying a DAF-16::green fluorescent protein (GFP) fusion [*daf-16* (*mu86*) + P*daf-16*::DAF-16::GFP) was generously provided by Daniel Kalman. DAF-16::GFP worms were cultivated, and synchronized cultures were raised to adulthood, at 16°C using protocols described above. Synchronized, reproductively sterile adults were colonized as described above. GFP fluorescence in individual worms was read on a COPAS Biosorter after 24 h (day 1) and 72 h (day 3); worms were refed on day 2 according to standard protocols.

**Community analysis, fitting, and ordination.** Data analysis was performed in R. Prior to analysis, data were filtered to remove individual worms with <100 CFU/worm (*daf-2*) or <1,000 CFU/worm (all other worm strains) to minimize errors from low counts on plates. Bacterial count data were log transformed and standardized using the function *decostand* in *vegan* (62), and principal-component analysis was performed using *PCA* from *FactoMineR* (63). Bray-Curtis distances were calculated from log-transformed data using *vegdist*, and Shannon diversity was obtained using the function *diversity* in *vegan*.

Dirichlet-multinomial fitting and simulations were performed using the package *dirmult* (64). Raw data were fitted using the core function *dirmult*, and simulated data were generated using *simpop* with

**TABLE 2** Dirichlet-multinomial maximum-likelihood fit parameters for N2, *daf-16* mutant, and *daf-2* mutant multispecies colonization data

| Strain or mutation (*n*) | Value for: | | | | | | | | Theta | logLik value |
|---|---|---|---|---|---|---|---|---|---|---|
| | pi:MYb120 | pi:MYb181 | pi:MYb238 | pi:MYb27 | pi:MYb45 | pi:MYb53 | pi:MYb56 | pi:MYb71 | | |
| N2 (164) | 0.0096 | 0.0015 | 0.0384 | 0.0866 | 0.029 | 0.0828 | 0.0903 | 0.6619 | 0.1698 | −2.20E + 06 |
| *daf-16* (100) | 0.0063 | 0.004 | 0.0594 | 0.2409 | 0.0237 | 0.0738 | 0.0592 | 0.5327 | 0.1597 | −2.21E + 06 |
| *daf-2* (98) | 0.0006 | NA | 0.0616 | 0.044 | 0.0133 | 0.1099 | 0.0719 | 0.6986 | 0.2524 | −1.03E + 05 |

the resulting parameters (Table 2). To generate simulated data sets for each host lineage, *simpop* was executed with host-specific parameters and a sample size drawn from the vector of total CFU per milliliter for that host; each sample in the original data set is simulated this way, resulting in a synthetic data set containing the same number of individual hosts with the same CFU/ml as in the original data. In these simulations, 10,000 data sets were simulated for each host lineage. Spearman correlations were calculated for each real and simulated data set using *rcorr* from the package *Hmisc* (65).

Data for the drop-down assays (Fig. 5) were further filtered before analysis to remove data for MYb181 bacteria from *daf-2* (no counts of this species) and N2 (nonzero counts in 10 of 156 worms), as inclusion of this species resulted in overweighting of rare MYb181 presence. Communities from *daf-16* worms showed higher prevalence of MYb181 (nonzero counts in 22 of 126 worms), and as removal of this species made no appreciable difference in the qualitative results of the ordination, it was not removed.

## SUPPLEMENTAL MATERIAL

Supplemental material is available online only.
**DATA SET S1**, XLSX file, 0.1 MB.
**FIG S1**, TIF file, 0.8 MB.
**FIG S2**, TIF file, 0.2 MB.
**FIG S3**, TIF file, 0.1 MB.
**FIG S4**, TIF file, 0.1 MB.
**FIG S5**, TIF file, 0.2 MB.
**FIG S6**, TIF file, 0.3 MB.
**FIG S7**, TIF file, 1.9 MB.
**FIG S8**, TIF file, 0.1 MB.
**TABLE S1**, XLSX file, 0.01 MB.

## ACKNOWLEDGMENTS

This research received no specific grant from any funding agency in the public, commercial, or not-for-profit sectors. This work was supported by funds provided by Emory College.

We thank Daniel Kalman and Guy Benian at Emory for generously providing us with worm strains.

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
