## [Reviewer comments · mSystems]

Host immunity alters community ecology and stability of the microbiome in a *C. elegans* model

Megan Taylor and Nic Vega

Corresponding Author(s): Nic Vega, Emory University

Review Timeline:

Submission Date:	July 8, 2020
Editorial Decision:	August 15, 2020
Revision Received:	December 2, 2020
Editorial Decision:	January 6, 2021
Revision Received:	February 15, 2021
Accepted:	March 17, 2021

Editor: John Rawls

Reviewer(s): Disclosure of reviewer identity is with reference to reviewer comments included in decision letter(s). The following individuals involved in review of your submission have agreed to reveal their identity: Julia Johnke (Reviewer #2)

Transaction Report:

DOI: <https://doi.org/10.1128/mSystems.00608-20>

August 15, 2020

Prof. Nic M Vega
Emory University
Biology
1510 Clifton Road
Atlanta, GA 30322

Re: mSystems00608-20 (Host immunity alters successional ecology and stability of the microbiome in a *C. elegans* model)

Dear Prof. Nic M Vega:

I have received three reviews of your manuscript. All found the manuscript to be potentially suitable for mSystems, but raised significant concerns and questions that will need to be addressed. In particular, Reviewer 2's concerns about appropriately controlling for potential developmental timing effects in *daf-2* and other mutants must be addressed fully.

Below you will find the comments of the reviewers.

To submit your modified manuscript, log onto the eJP submission site at <https://msystems.msubmit.net/cgi-bin/main.plex>. If you cannot remember your password, click the "Can't remember your password?" link and follow the instructions on the screen. Go to Author Tasks and click the appropriate manuscript title to begin the resubmission process. The information that you entered when you first submitted the paper will be displayed. Please update the information as necessary. Provide (1) point-by-point responses to the issues raised by the reviewers as file type "Response to Reviewers," not in your cover letter, and (2) a PDF file that indicates the changes from the original submission (by highlighting or underlining the changes) as file type "Marked Up Manuscript - For Review Only."

Due to the SARS-CoV-2 pandemic, our typical 60 day deadline for revisions will not be applied. I hope that you will be able to submit a revised manuscript soon, but want to reassure you that the journal will be flexible in terms of timing, particularly if experimental revisions are needed. When you are ready to resubmit, please know that our staff and Editors are working remotely and handling submissions without delay. If you do not wish to modify the manuscript and prefer to submit it to another journal, please notify me of your decision immediately so that the manuscript may be formally withdrawn from consideration by mSystems.

To avoid unnecessary delay in publication should your modified manuscript be accepted, it is important that all elements you upload meet the technical requirements for production. I strongly recommend that you check your digital images using the Rapid Inspector tool at <http://rapidinspector.cadmus.com/RapidInspector/zmw/>.

Sincerely,

John Rawls

Editor, mSystems

Journals Department
Reviewer comments:

Reviewer #1 (Comments for the Author):

In addressing the causes for inter-individual variation in microbiota composition, the authors take a quantitative approach to evaluate the roles of host metabolic state and immune system status versus the role of inter-species interactions among gut colonizers in affecting the composition of the gut microbiota. The study utilizes *C. elegans* as a model host, employing several mutants, and follows colonization from an eight-species bacterial community, members of which can be distinguished based on colony appearance and their colonization evaluated using CFU counts. The authors focus on the roles of the insulin signaling pathway, with mutants for DAF-16 - a central immune regulator, and for DAF-2, the insulin receptor, its negative regulator. They conclude that establishment of the gut microbiota and its ecological succession deviate from the neutral model. They show that host immunity imposes a constraint on bacterial colonization and on inter-species interactions, but that given this constraint, inter-species interactions, through ecological succession, play an important role in determining gut microbiota composition.

It is an interesting paper taking advantage of the authors' skills in quantitative analyses to extract useful information from a set of relatively straightforward experiments, suggesting new directions in utilizing a popular model organism to address fundamental questions in microbiome research. This study can be quite useful, but some glossing over potential caveats, unclarity in describing pivotal hypotheses (leading to results in Fig. 5), and over-interpretation of correlations as ecological succession dynamics weakens its conclusions. The authors should justify their choice of bacterial community, demonstrating that colonies could be efficiently distinguished in a mixed culture; consider inclusion a *daf-2;daf-16* double mutant control to enable focusing on DAF-16-dependent contributions of insulin signaling, which has broad effects on metabolism; acknowledge drawbacks relevant to some of the other mutants (e.g. VHP-1); and clearly distinguish between correlations between microbiota size and diversity in one time point and ecological succession along several

time points. This may require toning down of conclusions, but could overall strengthen the paper. Detailed comments are shown below:

1. Methods: The synthetic microbiota in use is made of eight strains chose solely based on their colony appearance to enable easy scoring. One would expect that their interactions represent the most generalized ones possible, i.e. competition for resources (with this in mind, the extent of positive interactions identified in Fig. 4 is surprising; what kind of interactions could they represent?). Altogether, it should be acknowledged that this community is probably not quite representative of the richer milieu of interactions between bacteria and their hosts in a natural context.

To provide sound basis for all analyses, the authors need to better demonstrate that the choice of bacteria was working. Fig. S1 demonstrates how different colonies of the different community members look when cultured each on its own. However, this is not quite convincing. How, for example, can one count colonies of Myb56? These bacteria seem to give rise to a diffused lawn. Furthermore, in a mixed culture, competition may affect the size of adjacent colonies, as well as their color, hindering proper identification (Myb45, Myb120, and Myb181 might be indistinguishable). At the very least the authors should demonstrate easy distinction on a mixed culture plate.

2. Claims about differences between microbiota composition in PCoA graphs should be supported by statistics, typically PERMANOVA, or ANOSIM, and would be easier to discern by including ellipses for the center of mass, with rays to the different microbiotas, as commonly done in such figures. Any thoughts about the nature of these differences?

3. Ecological succession in the worm gut and the rules governing it should be cautiously interpreted. Yes, the idea that diversity decreases with community size makes sense, and has support in the literature. True, the authors identify a supporting correlation in D4 microbiotas. However, this is one time point, and not ecological succession per se. Furthermore, when the authors look how this trend holds when considering real ecological succession, i.e. D2, 4 and 6, the hypothesized rule is not well supported (at D6). All this should be acknowledged (better than as it is now hand-waved away on line 91) and be made clear rather than trying to present a stronger than possible support for rules governing ecological succession.

4. The competing hypotheses leading to the drop-out experiments shown in Fig. 5, (around line 165) are not clear, and the interpretation of the results of these experiments is not clear. This is an important part of the paper and should be well explained.

5. To present the biological data, which is later-on abstracted in PCoA graphs, the authors should include in Fig. 1B the bar graphs representing microbiotas of all mutants.

6. The authors use a 15 min incubation with bleach to remove external contamination from worms. The bleach concentration is relatively low but the incubation is long. This is performed at 4C to stop intestinal peristalsis, presumably preventing bleach from entering the gut and affecting colonizing bacteria, but can they show that this works? Otherwise, microbiota composition might be biased by differential resistance to the leaking bleach.

7. Mutants in use:

- DAF-2 is not only a repressor of DAF-16, although this is one of its more important and characterized roles. It affects metabolism as well, and this should be considered. Including experiments in *daf-2;daf-16* double mutants could serve as an important control for DAF-16-

specific contributions of DAF-2. It should further be acknowledged that DAF-16 is not simply an immune regulator, but a regulator of many other stress responses, including oxidative stress, and xenobiotics. With that in mind, it's also worth acknowledging that DBL-1 signaling is also important to development, not only immunity.

- Most studies on the role of vhp-1 were performed with RNAi. Null vhp-1 mutants are inviable. The vhp-1(sa366) strain is viable due to a weaker phenotype, and is probably a hypomorph, so it's not clear to what extent it leads to p38 over-activation. This should be acknowledged.

While it's understood that glp-4 mutants should serve as a control for pmk-1 mutants that share the same glp-4 background, it should be acknowledged that glp-4 disruption further affects protein translation, which may have additional effects. Furthermore, the vhp-1 mutants in use do not share this background, which further complicates matters.

Minor points.

1. Line 47: Saying that grinding mutants were not examined before for effects on microbiome composition is inaccurate. See Berg et al. 2019, for microbiota composition in tnt-3 mutants. The results here should be compared to the previous ones.

2. Line 36, should be Fig. 1A.

3. Where are Figs 2E and 2F (line 63)?

4. To improve the accessibility for the reader it would be helpful to split the results part into paragraphs with meaningful headers.

5. Generally, please italicize gene names and species names also in graphs (e.g. Tab. 1, Supp Fig. 3, 4, 5...), and label axes understandably (e.g. x-axis of Supp. Fig. 10).

6. lines 110-115: A Spearman correlation coefficient was calculated for each worm strain but also bacterial strain (Supp. Fig. 6) but it is not clear which variables are correlated with each other. Please elaborate.

7. line 281: Please explain what is AXN.

8. Fig. 3: It would be helpful to provide the correlation coefficient in the plot or the Fig. legend.

9. Fig. 2, Supp. Fig. 3B: The colors of worm strains db1-1 and ctIs40 are too similar so that the lack of change in microbiome composition between ctIs40 and N2 (as stated in lines 61/62) is not apparent in the graph.

10. Supp. Fig. 1: Please confirm that pictures were taken with the same magnification, or add a scale bar.

11. Supp. Fig. 2: Please explain the ten-worm moving window.

Propagule pressure is likely not familiar to many. Adding an explanation would be useful.

Reviewer #2 (Comments for the Author):

Comments to the author:

In this manuscript the authors show that host immunity has a strong effect on microbiome community composition and structure (in particular the interactions between the microbiota). Even though this is not the first study that shows the influence of *C. elegans* genetics on the microbiome, it is the first that shows a direct effect on bacteria-bacteria interactions and the effect of a perturbation on community assembly.

Specific comments:

Line 33: the authors claim that the community was selected to "represent the overall diversity of bacterial taxa in these samples [the *C. elegans* microbiome]", however, some of the "core" *C. elegans* microbiota are missing (e.g. *Pseudomonas*) and the diversity of natural *C. elegans* is much higher. I understand that the selection was also based on colony morphology, which makes sense. I

would still suggest to re-phrase the sentence.

Additionally, in Fig. S1 the picture of MYb56 does not support the claim that counting of individual colonies was possible. It would be good to see one that contains separate colonies.

Fig. 1c: in this PCA the axis are labeled as Dim 1 and Dim 2, later they are labeled as PC1 and PC2. I would suggest to use continuously the same names.

Line 36-37: I wonder how the frequency distribution of total colonization looked like. This would help to interpret Fig. S2. Did all these worms stem from the same well of the experimental plate or were the sampled from replicate wells (this should be also addressed in the methods section)?

Line 40: "Community composition changes over the course of colonization" should be re-phrased as individual worms were sampled (as stated in the Fig. legend).

Fig. 2: some of the colors are hard to tell apart, especially the yellow and brown in 2c. The same is true for Fig. 3b, 5a, f, k, d, i, n, and Fig. S9.

Fig. S3: Could you please state the N for each of the mutants?

Line 51: "...similar to N2 communities of the same size". Does Fig. S3 only show the community composition of worms that shared the same bacterial load?

Line 63: should be Fig. 2c

Line 64: should be Fig. 2d

Line 93: should be Fig. 3e

Line 93-94: but the Shannon index is not decreasing. Here, it would be nice to have a graph that shows the differences in Shannon index for the different mutants. Fig. 3e should be 3f. It seem rather that the bacterial load is lower.

Fig. 3: In the legend, (E) appears two times. E and F show a correlation between bacterial load and Shannon index. I wonder if the beta-diversity also changes over time (as it was shown for N2). For D, E, F: it looks as if there are differences in bacterial load between the mutants (especially daf-16), but it is hard to grasp from the type of graph. Is this difference statistically significant?

Fig. 4: what is the difference between Fig. 4 a-c and Fig. S6 b, d, f? Also, I find it easier to grasp the differences in the histograms shown in Fig. S6 than those in Fig. 4.

Line 250 ff: Please clarify if the developmental timing of the different mutants was controlled or monitored. Did they all develop in the same speed? If not, this, and how this might affect the results should be discussed, especially as daf-2 mutants usually develop at a slower rate than WT N2. This would then switch the "mid succession" window that was only identified for N2.

Line 297: "constant cell densities" sounds as if the density was kept constant during the experiment, which, I think was not the case. This should be clarified.

Line 301 ff: Did you also perform biological replicates (in the sense that multiple wells contained the same mutant and community) or did all worms that were analyzed per mutant stem from the same well?

Line 285 ff: As far as I understand all worms (with the exception of AU37 and glp-4) were constantly kept at 25{degree sign}C. Can you please clarify how this effected the daf-2 (e1370) mutant as the dauer phenotype is temperature sensitive? This mutant develops into dauers when kept at 25{degree sign}C. Does that mean that dauers were analyzed? This should affect the colonization and would make it hard to compare it to N2 adults.

Line 338 ff: I understand a PCA was based on Bray-Curtis distances? PCA is generally based on the Euclidean distance, but PCoAs can be used with Bray-Curtis distances, as far as I understand. The latter would additionally allow for statistical testing of a clustering of the groups of interest via perMANOVA, which would strengthen the results section.

Reviewer #3 (Comments for the Author):

In this manuscript by Taylor & Vega, the authors explore the relationships between host genotype

and microbiome. This study was carried out under controlled settings, using the microbiome of 8 species, which representing the diversity of the full microbiome of *C. elegans*. This paper adds to the existing literature and will be of interest to multiple labs that study *C. elegans* ecology, host-microbe, or host-pathogen interactions.

The authors combined canonical colonization assays with ecological approaches. For example they used the Shannon diversity index and also analyzed relationship between bacteria within the host, and the relationship between bacteria and host. The paper is generally well-written and easy to follow. The Analysis portion of Methods section is described sufficiently well for this process to be recapitulated by other labs.

I only have one major question: Of the three pathways tested (PMK-1 p38/MAPK, TGF- β , and Insulin signaling), do the authors know that any of these were active under the conditions tested? If the pathway is not active, then it is likely to expect that its knockout will have little effect on microbial composition. However, overexpression could easily have unexpected (and potentially artefactual) consequences. Fluorescent reporters for downstream genes (or qPCR of such genes), or direct DAF-16::GFP fusions may allow this question to be answered easily and provide more mechanistic connections between the presence / absence of the effect for a given pathway.

Minor points:

Check the manuscript for the consistent and standard use of the nomenclature: genotypes in Table 1 or Fig 2 are not italicized. In Materials and Methods, one strain is referred is AU37, and one next to it as *glp-4*, instead of SS104. Similar mixing is in Fig 2. *C. elegans* needs to be italicized (e.g. Fig 2 legend).

For Fig. 2, can statistics be generated to compare communities in different mutants?

Of the three pathways tested (PMK-1 p38/MAPK, TGF- β , and Insulin signaling), which were active under the conditions tested? If pathway is not active, then it is likely to expect that its knockout will have little effect on microbial composition, but overexpression may have consequences (possibly, no physiological). A fluorescent reporter for downstream genes (or qPCR of such genes), or direct DAF-16::GFP fusion may allow to answer this question easily, providing more mechanistic connection between the presence / absence of the effect for a given pathway.

Reviewer comments:

Reviewer #1 (Comments for the Author):

In addressing the causes for inter-individual variation in microbiota composition, the authors take a quantitative approach to evaluate the roles of host metabolic state and immune system status versus the role of inter-species interactions among gut colonizers in affecting the composition of the gut microbiota. The study utilizes *C. elegans* as a model host, employing several mutants, and follows colonization from an eight-species bacterial community, members of which can be distinguished based on colony appearance and their colonization evaluated using CFU counts. The authors focus on the roles of the insulin signaling pathway, with mutants for DAF-16 - a central immune regulator, and for DAF-2, the insulin receptor, its negative regulator. They conclude that establishment of the gut microbiota and its ecological succession deviate from the neutral model. They show that host immunity imposes a constraint on bacterial colonization and on inter-species interactions, but that given this constraint, inter-species interactions, through ecological succession, play an important role in determining gut microbiota composition.

It is an interesting paper taking advantage of the authors' skills in quantitative analyses to extract useful information from a set of relatively straightforward experiments, suggesting new directions in utilizing a popular model organism to address fundamental questions in microbiome research. This study can be quite useful, but some glossing over potential caveats, unclarity in describing pivotal hypotheses (leading to results in Fig. 5), and over-interpretation of correlations as ecological succession dynamics weakens its conclusions. The authors should justify their choice of bacterial community, demonstrating that colonies could be efficiently distinguished in a mixed culture; consider inclusion a *daf-2;daf-16* double mutant control to enable focusing on DAF-16-dependent contributions of insulin signaling, which has broad effects on metabolism; acknowledge drawbacks relevant to some of the other mutants (e.g. VHP-1); and clearly distinguish between correlations between microbiota size and diversity in one time point and ecological succession along several time points. This may require toning down of conclusions, but could overall strengthen the paper. Detailed comments are shown below:

1. Methods: The synthetic microbiota in use is made of eight strains chose solely based on their colony appearance to enable easy scoring. One would expect that their interactions represent the most generalized ones possible, i.e. competition for resources (with this in mind, the extent of positive interactions identified in Fig. 4 is surprising; what kind of interactions could they represent?). Altogether, it should be acknowledged that this community is probably not quite representative of the richer milieu of interactions between bacteria and their hosts in a natural context.

We thank the reviewer for raising this point, as we believe this is an important consideration with regard to interpretation of these results. The reviewer is correct that this community is minimal, containing only eight strains, and for that reason extrapolation to larger communities should be done with caution. Additionally, this minimal community was selected from strains representing the worm native microbiome in part based on practical considerations. However, these are strains isolated from a study of the wild native microbiome of *C. elegans*, and may (or may not) have an

evolutionary history that would modify their interactions with one another and/or with the nematode host.

Further, taxonomic (and therefore presumably functional) diversity was another primary consideration in selecting these strains from the original bank of isolates. A recent publication by the lab where these species were isolated has characterized the functional diversity in these bacteria, indicating that metabolic competencies are predicted to be closely tied to taxonomy in these strains, and that higher-order interactions such as cross-feeding are observed, although the latter was tested in only a small subset of strains for which metabolic models could be constructed (Zimmerman et al. 2020, ref 21). We are in the process of characterizing the interactions among these species in more detail, and our results thus far are broadly consistent with expectations from Zimmerman et al. Based on these results, as well as our own experience with these strains, we believe that this minimal community, while small, is functionally diverse, and contains a sufficiently interesting diversity of interactions.

The text has been modified to indicate “These bacterial strains represent a taxonomically and functionally diverse subset of isolates from a wild *C. elegans* microbiome” and to cite the indicated reference (**line 33**). In the Discussion (**lines 264-268**) we have added the text “We chose to use a minimal eight-species community for tractability, and to select our community to represent a taxonomically and functionally diverse subset of the native host microbiome to allow for a range of possible interactions between species (20, 21). However, the small size of this community means that extrapolation to larger microbiota should be done with caution.”

In Figure 4, it is important to keep in mind that we are working with count data and not relative abundance. Positive correlations in these data do not necessarily reflect real positive interactions among bacteria – non-interacting or neutrally interacting bacteria can easily show these patterns when the number of bacteria per host varies across individuals. This is why we are able to get strong (but spurious) positive correlations in the simulated data in Figure S9 (formerly S8), despite the fact that the DMN model does not contain true positive *or* negative interactions between species. We have edited the text to reinforce this point, stating (**line 130-132**): “Note that positive correlations in these data need not reflect true positive interactions between species; a positive relationship between counts of two bacteria can simply reflect common species being common together, when the number of total bacteria varies among samples as it does in these data.”

To provide sound basis for all analyses, the authors need to better demonstrate that the choice of bacteria was working. Fig. S1 demonstrates how different colonies of the different community members look when cultured each on its own. However, this is not quite convincing. How, for example, can one count colonies of Myb56? These bacteria seem to give rise to a diffused lawn. Furthermore, in a mixed culture, competition may affect the size of adjacent colonies, as well as their color, hindering proper identification (Myb45, Myb120, and Myb181 might be indistinguishable). At the very least the authors should demonstrate easy distinction on a mixed culture plate.

We have added additional images to Supplementary Figure 1, demonstrating that colony morphologies remain distinct and are easily identified on community plates.

2. Claims about differences between microbiota composition in PCoA graphs should be supported by statistics, typically PERMANOVA, or ANOSIM, and would be easier to discern by including ellipses for the center of mass, with rays to the different microbiotas, as commonly done in such figures. Any thoughts about the nature of these differences?

We have updated the plots in Figure 2 to include ellipses for center of mass, and the results of statistical testing are described briefly in the text (**lines 50-51**): “All worm mutants were colonized under the same conditions used for N2 (above), and preliminary testing indicated differences between host strains (ANOSIM based on Bray-Curtis differences, 9999 permutations, $R=0.27$, $p<0.0001$; perMANOVA, 999 permutations, $F=54.624$, $p=0.001$).”

We hold that a convincing argument against the use of starburst plots for microbiome ordination data was made in Knights et al. 2014 (Rethinking “Enterotypes”, *Cell Host Microbe* 16(4):433-437), who indicated that these plots can suggest patterns where none exist. Further, due to the very large number of points in this data set, we believe that the extra plot components added by the starburst would make these plots excessively crowded and more difficult to read.

3. Ecological succession in the worm gut and the rules governing it should be cautiously interpreted. Yes, the idea that diversity decreases with community size makes sense, and has support in the literature. True, the authors identify a supporting correlation in D4 microbiotas. However, this is one time point, and not ecological succession per se. Furthermore, when the authors look how this trend holds when considering real ecological succession, i.e. D2, 4 and 6, the hypothesized rule is not well supported (at D6). All this should be acknowledged (better than as it is now hand-waved away on line 91) and be made clear rather than trying to present a stronger than possible support for rules governing ecological succession.

We thank the reviewer for pointing out our lack of clarity on this point – it was not our intention to actively support any particular theory of ecological succession with these data, merely to point out that our data are not inconsistent with the theoretical rule described here. As the reviewer states, these experiments were not designed as a test of diversity-size relationships and provide at best indirect (and host genotype-specific) evidence for any such relationship. For these data, the more important point is the difference in successional pattern in the different host genotypes. **We have therefore removed the statement in question.**

4. The competing hypotheses leading to the drop-out experiments shown in Fig. 5, (around line 165) are not clear, and the interpretation of the results of these experiments is not clear. This is an important part of the paper and should be well explained.

We concur with the reviewer on this point. The competing hypotheses underlying these experiments are rather technical and can be difficult to explain in a way that is clear to a broad audience. We have re-written the section in question to make the competing hypotheses more clear, as follows (**lines 180-195**):

“We expect this perturbation to have specific effects if there are substantial priority effects (where the early composition of the community affects the species-specific probability of success of later colonists) mediated by inter-species interactions. We should see very little effect in a highly stringent host (*daf-2*) where environmental filtering dominates, as we expect bacterial species to colonize based on ability to survive this environment rather than on ability to interact with other colonizers. Conversely, this perturbation may have a large effect in an uncontrolled host (*daf-16*) where interactions between bacteria, rather than interactions with the host, are the (theorized) dominant force driving community assembly.

Other outcomes are possible. If priority effects are mediated through changes in the host state rather than through interactions between bacterial species, this perturbation should produce smaller changes in environments where host control is of less relative importance (N2, *daf-16*) than in environments where the host is the dominant factor (*daf-2*). Alternately, if these systems are not driven by priority effects, the small perturbation introduced in these experiments should not alter community composition.”

5. To present the biological data, which is later-on abstracted in PCoA graphs, the authors should include in Fig. 1B the bar graphs representing microbiotas of all mutants.

These data are now presented as **Supplementary Figure 3A**.

6. The authors use a 15 min incubation with bleach to remove external contamination from worms. The bleach concentration is relatively low but the incubation is long. This is performed at 4C to stop intestinal peristalsis, presumably preventing bleach from entering the gut and affecting colonizing bacteria, but can they show that this works? Otherwise, microbiota composition might be biased by differential resistance to the leaking bleach.

We have validated the bleach protocol using highly sensitive *E. coli* DH5 α as a colonist of the worm gut, demonstrating that the protocol used here substantially reduces external bacteria without affecting intestinal counts of this easily killed bacteria. **These data are now presented in supplementary Figure S10D**. More anecdotally, we have used this bleaching protocol to sanitize larvae from contaminated plates; early instar (L1-L2) larvae do not transfer the contaminant after bleaching, while adult worms do, suggesting that the protocol works best for cleaning worms too small to carry contaminating microbes effectively in the gut.

7. Mutants in use:

DAF-2 is not only a repressor of DAF-16, although this is one of its more important and characterized roles. It affects metabolism as well, and this should be considered. Including experiments in *daf-2;daf-16* double mutants could serve as an important control for DAF-16-specific contributions of DAF-2. It should further be acknowledged that DAF-16 is not simply an immune regulator, but a regulator of many other stress responses, including oxidative stress, and xenobiotics.

We thank the reviewer for raising this point, and we concur that the double mutant control provides important additional information about the specific contributions of DAF-16 to

microbial community assembly. We have added a *daf-2(e1370);daf-16(mu86)* double mutant to our intestinal community data set. As the double mutant is sterile, it was necessary to make use of a mutant carrying an extrachromosomal array for propagation (CGC CF1449). The array is extremely unstable; non-GFP non-Rol animals, which have lost the extrachromosomal array, were very abundant in our samples and were easily sorted for single worm digests. As the *daf-2* and double mutant were best grown at 16C, we grew all comparison strains (N2, *daf-16*) to adulthood at 16C as well to provide appropriate controls, which provided further information about the likely role of *daf-16* in regulation of host-microbiome interactions in this system. **This information has been added to Methods and Table 1, and these data have been added to Figure 2 and Figure S4.**

We have altered the text in the Discussion to indicate the broad role of DAF-16 (**line 288**): “This pathway has been implicated in a broad range of stress responses (45, 46) including oxidative stress and xenobiotics in addition to response to bacterial pathogens...”

With that in mind, it's also worth acknowledging that DBL-1 signaling is also important to development, not only immunity.

Most studies on the role of *vhp-1* were performed with RNAi. Null *vhp-1* mutants are inviable. The *vhp-1(sa366)* strain is viable due to a weaker phenotype, and is probably a hypomorph, so it's not clear to what extent it leads to p38 over-activation. This should be acknowledged.

While it's understood that *glp-4* mutants should serve as a control for *pmk-1* mutants that share the same *glp-4* background, it should be acknowledged that *glp-4* disruption further affects protein translation, which may have additional effects. Furthermore, the *vhp-1* mutants in use do not share this background, which further complicates matters.

We have added text to the Discussion to clarify the limitations of the mutants used here (**lines 279-286**): “Interpretation of these results is complicated by the pleotropic nature of worm genetics. For example, DBL-1 is important in development as well as immunity (45). Disruption of *glp-4* is known to have broad effects, including changes in protein translation (46); the *glp-4* mutant was used here as a control for AU37, and the *vhp-1(sa66)* mutant used here for up-regulation of p38 does not have the *glp-4* mutation. Further, this *vhp-1* allele is probably hypomorphic (null alleles are lethal (47, 48)) with respect to p38 regulation. While our results suggest a complex role for innate immunity in regulation of the worm gut microbiome, further investigation of specific pathways is needed to disentangle immunity from other functions.”

Minor points.

1. Line 47: Saying that grinding mutants were not examined before for effects on microbiome composition is inaccurate. See Berg et al. 2019, for microbiota composition in *tnt-3* mutants. The results here should be compared to the previous ones.

We thank the reviewer for drawing our attention to this point of confusion. Here we intended to indicate that these specific mutants, which are mechanically different from those used in Berg et

al, had not been explored. However, as the conclusions are unchanged, we have rephrased the section in question to indicate that our results are consistent with those observed by Berg et al. **(line 56)**: “Increased permissiveness of the defective grinder did not substantially affect community assembly, consistent with previous results (24).”

2. Line 36, should be Fig. 1A.

3. Where are Figs 2E and 2F (line 63)?

We have corrected the figure references.

4. To improve the accessibility for the reader it would be helpful to split the results part into paragraphs with meaningful headers.

We have inserted headers into the Results section to break up the text into sections: *Effects of host genetics on microbiome assembly*; *DAF-2 signaling alters host control of microbiome assembly*; *DAF-2 signaling alters sensitivity of the microbiome to changes in colonization*.

5. Generally, please italicize gene names and species names also in graphs (e.g. Tab. 1, Supp Fig. 3, 4, 5...), and label axes understandably (e.g. x-axis of Supp. Fig. 10).

We have edited the indicated objects and legends to ensure correctness.

6. lines 110-115: A Spearman correlation coefficient was calculated for each worm strain but also bacterial strain (Supp. Fig. 6) but it is not clear which variables are correlated with each other. Please elaborate.

Spearman correlation coefficients were calculated for all pairs of bacterial species within each of the three host strains examined (N2, *daf-2*, *daf-16*); no correlations were calculated across host strains. We have edited the indicated text for clarity (**now line 124**): “To test this hypothesis, we first calculated the Spearman correlations for pairs of bacterial species within each of the N2, *daf-16*, and *daf-2* hosts”.

7. line 281: Please explain what is AXN.

An explanation, and the original citation, have been provided.

8. Fig. 3: It would be helpful to provide the correlation coefficient in the plot or the Fig. legend.

The summaries for all regressions are given in **Table S1**, as we now indicate in the figure legend, and we have moved the plot describing the linear fit to N2 data into Figure 3 (now Figure 3A).. Adding the entire set of coefficients and labeling these coefficients appropriately would produce

a significant amount of clutter due to the relatively large number of individual regressions in this set of plots. As these fits are merely descriptive and used here only to make confirmatory comparisons across host genotypes, we do not feel that the gain in information is worth the loss of clarity in this case.

9. Fig. 2, Supp. Fig. 3B: The colors of worm strains *dbl-1* and *ctIs40* are too similar so that the lack of change in microbiome composition between *ctIs40* and N2 (as stated in lines 61/62) is not apparent in the graph.

We have modified the color scheme to increase contrast among these worm strains.

10. Supp. Fig. 1: Please confirm that pictures were taken with the same magnification, or add a scale bar.

The images were taken on a handheld camera without scale bar capacity. **We have added multi-species plates to Figure S1** to provide context for how these colonies appear in proximity.

11. Supp. Fig. 2: Please explain the ten-worm moving window.

We have added explanatory text to the figure legend: "Running average of relative abundance data, calculated using a ten-worm moving window (average over all consecutive sets of ten worms in the ordered data set in A)."

Propagule pressure is likely not familiar to many. Adding an explanation would be useful.

We have added the following explanation to the text for clarity (**line 175**): "reduced propagule pressure (number of individuals of this species introduced per event, or per unit time)"

Reviewer #2 (Comments for the Author):

Comments to the author:

In this manuscript the authors show that host immunity has a strong effect on microbiome community composition and structure (in particular the interactions between the microbiota). Even though this is not the first study that shows the influence of *C. elegans* genetics on the microbiome, it is the first that shows a direct effect on bacteria-bacteria interactions and the effect of a perturbation on community assembly.

Specific comments:

Line 33: the authors claim that the community was selected to "represent the overall diversity of bacterial taxa in these samples [the *C. elegans* microbiome]", however, some of the "core" *C. elegans* microbiota are missing (e.g. *Pseudomonas*) and the diversity of natural *C. elegans* is

much higher. I understand that the selection was also based on colony morphology, which makes sense. I would still suggest to re-phrase the sentence.

We thank the reviewer for raising this point; the reviews are in accord on this, that it is necessary for us to clarify the grounds on which these bacteria were selected and to emphasize the functional diversity that is conserved by our selection process. We have re-phrased the sentence in question to indicate: “These bacterial strains represent a taxonomically and functionally diverse subset of isolates from a wild *C. elegans* microbiome (20, 21) (**Methods**). Each possessed a unique colony morphology when co-cultured on agar plates, allowing CFU counts for each species to be taken from mixed communities (**Fig S1**).”

Additionally, in Fig. S1 the picture of MYb56 does not support the claim that counting of individual colonies was possible. It would be good to see one that contains separate colonies.

We have provided additional images in Figure S1 to clarify the morphology of these bacteria and indicate the conservation of these distinct morphologies on community plates.

Fig. 1c: in this PCA the axis are labeled as Dim 1 and Dim 2, later they are labeled as PC1 and PC2. I would suggest to use continuously the same names.

We have changed the axis labels on all figures to consistently read “PC1” and “PC2”.

Line 36-37: I wonder how the frequency distribution of total colonization looked like. This would help to interpret Fig. S2. Did all these worms stem from the same well of the experimental plate or were the sampled from replicate wells (this should be also addressed in the methods section)?

We have added a histogram of the total CFU/worm data to Figure S2 to aid in interpretation of these data. There is clearly heterogeneity in colonization intensity, but the heterogeneity is not unstructured; a more complete description of the between-individual variation is beyond the scope of the present manuscript.

In this data set, N2 worms were sampled from twelve independent experiments conducted over the course of roughly a year. Each individual experiment contains data for 12-36 individual N2 worms taken from a single well. Other host strains follow a similar pattern and are represented by 2+ independent experiments where 12-36 individual worms were digested per strain. This information has been added to the **Figure 1 legend** and to the Methods (**lines 378-380**) to clarify the structure of the data.

Line 40: "Community composition changes over the course of colonization" should be re-phrased as individual worms were sampled (as stated in the Fig. legend).

The reviewer is correct that these data do not represent an auto-correlated time series, and we

concur that this is an important point with regard to these data. We have corrected the text to read **(line 43)**: “Community composition differs in worms sampled at different time points in colonization (Fig. 1C)”

Fig. 2: some of the colors are hard to tell apart, especially the yellow and brown in 2c. The same is true for Fig. 3b, 5a, f, k, d, i, n, and Fig. S9.

We have modified the color schemes in these figures to increase contrast among these worm strains.

Fig. S3: Could you please state the N for each of the mutants?

Sample size information for each of the mutants has been added to the figure legend **(now Figure S4)**.

Line 51: "...similar to N2 communities of the same size". Does Fig. S3 only show the community composition of worms that shared the same bacterial load?

The original statement should refer to the original figure S4 (now S5), showing the diversity-size relationships. However, in the interests of clarity, we have rephrased the indicated statement to read **(lines 54-56)** “While communities in the severe grinder mutant *phm-2* were large compared to N2 (median CFU/worm 36,190 vs 13,000), composition was within the range observed for N2 (Fig. 2A)”

Line 63: should be Fig. 2c

Line 64: should be Fig. 2d

Line 93: should be Fig. 3e

We have corrected the figure references as indicated.

Line 93-94: but the Shannon index is not decreasing. Here, it would be nice to have a graph that shows the differences in Shannon index for the different mutants. Fig. 3e should be 3f. It seems rather that the bacterial load is lower.

We thank the reviewer for bringing this to our attention; our original statement was inaccurate. We have re-stated the description in the text as **(lines 105-108)**: “*daf-16* hosts displayed large populations which continued to increase in size and diversity over the observed period (Fig. 3E, Fig. S6D-F), while *daf-2* hosts showed convergence to smaller microbiomes consisting mainly of three dominant bacteria (Fig. 3F, Fig S5G-I).”

Fig. 3: In the legend, (E) appears two times. E and F show a correlation between bacterial load and Shannon index. I wonder if the beta-diversity also changes over time (as it was shown for N2). For D, E, F: it looks as if there are differences in bacterial load between the mutants (especially *daf-16*), but it is hard to grasp from the type of graph. Is this difference statistically significant?

We have corrected the figure 3 legend.

The CFU/worm data in Figure S5 (**now Figure S6**; same data as summarized in Figure 3) show the trend in population size more clearly, as well as displaying the composition of these communities directly; we now refer the reader directly to this figure in the Figure 3 legend.

Additionally, we have added a violin plot of total CFU/worm counts to Figure S6. **The legend describes the results of statistical testing to compare total counts between groups, with p-value bins indicated on the graph.**

Fig. 4: what is the difference between Fig. 4 a-c and Fig. S6 b, d, f? Also, I find it easier to grasp the differences in the histograms shown in Fig. S6 than those in Fig. 4.

Only the programs used are different (Fig S6b,d,f were generated in Excel). Both use the same bin size, but make slightly different bin allocations; Excel takes bin numbers as upper levels, and R seems to use lower-level bins. The data are the same.

Line 250 ff: Please clarify if the developmental timing of the different mutants was controlled or monitored. Did they all develop in the same speed? If not, this, and how this might affect the results should be discussed, especially as *daf-2* mutants usually develop at a slower rate than WT N2. This would then switch the "mid succession" window that was only identified for N2.

The mutants do not all develop at the same speed. Part of troubleshooting these experiments was working out the timing of synchronizations. We sought to ensure that synchronized worms from different strains would reach adulthood at the same time so that they could then be colonized in parallel. Sometimes this required staggering synchronizations; in other cases, it was possible to group strains that grew similarly into experimental clusters. In the case of *daf-2*, for example, it was sometimes simplest to run replicates of this strain along with replicates of other slow strains such as *dec-1* (not shown – part of a separate project).

We have edited the Methods to indicate that this was the case (**lines 334-336**): “As some of the mutants used here develop to adulthood at different rates, care was taken to arrange and/or stagger synchronizations such that all strains within a given experiment reached adulthood at roughly the same time.”

Line 297: "constant cell densities" sounds as if the density was kept constant during the experiment, which, I think was not the case. This should be clarified.

The reviewer is correct; the bacterial metacommunities are started at a relatively high titer ($\sim 10^8$ CFU/mL), which helps to minimize change due to growth by limiting the number of bacterial generations in the liquid media, and are replaced every 48 hours to minimize change in the community and provide a continual source of live, fresh bacteria for colonization. However, there is some inevitable shift over time. We have replaced the word "constant" with "uniform" to indicate the initial conditions used.

Line 301 ff: Did you also perform biological replicates (in the sense that multiple wells contained the same mutant and community) or did all worms that were analyzed per mutant stem from the same well?

All worm strains in the intestinal community data set are represented by multiple (2+, usually 3+) biological replicates, conducted on separate days, with 12-36 worms taken per well per experiment depending on the total number of worm strains being processed in that experiment. (We use a 96-well mechanical disruption procedure, making it convenient to work in multiples of 12.) This is now reflected in the Methods (**lines 378-380**) and can be seen in the raw count data provided in Data Set S1.

Line 285 ff: As far as I understand all worms (with the exception of AU37 and *glp-4*) were constantly kept at 25{degree sign}C. Can you please clarify how this effected the *daf-2* (e1370) mutant as the dauer phenotype is temperature sensitive? This mutant develops into dauers when kept at 25{degree sign}C. Does that mean that dauers were analyzed? This should affect the colonization and would make it hard to compare it to N2 adults.

The *daf-2* strain was cultivated, and synchronized worms were raised to adulthood, at 16°C to prevent constitutive dauer formation; this is now reflected in the Methods (**line 331**, "Starved L1 larvae were transferred to 10cm NGM plates containing lawns of *E. coli pos-1* RNAi and incubated at 25C for 3 days (most) or 16C (*daf-2*; CF1449; N2 and *daf-16* when specifically stated) to produce reproductively sterile adults").

Figures 2 and S4 now contain a comparison of N2, *daf-2*, *daf-16*, and *daf-2; daf-16* adults raised to adulthood at 16°C. Briefly, it is apparent that the *daf-2* mutant exerts its effects on host-microbe interactions substantially through dysregulation of *daf-16*, and the effects of *daf-16* itself are dependent on the temperature at which larvae were raised.

Line 338 ff: I understand a PCA was based on Bray-Curtis distances? PCA is generally based on the Euclidean distance, but PCoAs can be used with Bray-Curtis distances, as far as I understand. The latter would additionally allow for statistical testing of a clustering of the groups of interest via perMANOVA, which would strengthen the results section.

The PCA is based on the log-transformed and standardized count data as indicated in the Methods. We initially used PCA here because, first, we could; these are non-sparse count data, suitable for Euclidean distance calculations after appropriate standardization. Second, use of PCA allowed us to establish the contribution of individual bacterial species to the ordination results; however, these results did not prove to be sufficiently interesting to justify exploring in the manuscript. Further, our choice of day 4 for sampling in the large data set meant that we were, with full forethought, not maximizing distances between host strains in these data; we expect false negatives in perMANOVA or equivalent tests, with respect to biological differences.

Although these experiments were not ideally designed for statistical tests of differences between groups, it is not incorrect to use this test on these data. **We have therefore edited the text to indicate the results of ANOSIM and perMANOVA testing based on host strain**, following calculation of the Bray-Curtis distances (**lines 50-51**): “All worm mutants were colonized under the same conditions used for N2 (above), and preliminary testing indicated differences between host strains (ANOSIM based on Bray-Curtis differences, 9999 permutations, $R=0.27$, $p<0.0001$; perMANOVA, 999 permutations, $F=54.624$ $p=0.001$).”

Reviewer #3 (Comments for the Author):

In this manuscript by Taylor & Vega, the authors explore the relationships between host genotype and microbiome. This study was carried out under controlled settings, using the microbiome of 8 species, which representing the diversity of the full microbiome of *C. elegans*. This paper adds to the existing literature and will be of interest to multiple labs that study *C. elegans* ecology, host-microbe, or host-pathogen interactions.

The authors combined canonical colonization assays with ecological approaches. For example they used the Shannon diversity index and also analyzed relationship between bacteria within the host, and the relationship between bacteria and host. The paper is generally well-written and easy to follow. The Analysis portion of Methods section is described sufficiently well for this process to be recapitulated by other labs.

I only have one major question: Of the three pathways tested (PMK-1 p38/MAPK, TGF- β , and Insulin signaling), do the authors know that any of these were active under the conditions tested? If the pathway is not active, then it is likely to expect that its knockout will have little effect on microbial composition. However, overexpression could easily have unexpected (and potentially artefactual) consequences. Fluorescent reporters for downstream genes (or qPCR of such genes), or direct DAF-16::GFP fusions may allow this question to be answered easily and provide more mechanistic connections between the presence / absence of the effect for a given pathway.

Minor points:

Check the manuscript for the consistent and standard use of the nomenclature: genotypes in

Table 1 or Fig 2 are not italicized. In Materials and Methods, one strain is referred to as AU37, and one next to it as *glp-4*, instead of SS104. Similar mixing is in Fig 2. *C. elegans* needs to be italicized (e.g. Fig 2 legend).

We have corrected the use of italics in figures, tables, and legends, and we have standardized our presentation of the strains in the Methods.

For Fig. 2, can statistics be generated to compare communities in different mutants?

We have edited the text to indicate the results of ANOSIM and perMANOVA testing based on host strain, following calculation of the Bray-Curtis distances (**lines 50-51**): “All worm mutants were colonized under the same conditions used for N2 (above), and preliminary testing indicated differences between host strains (ANOSIM based on Bray-Curtis differences, 9999 permutations, $R=0.27$, $p<0.0001$; perMANOVA, 999 permutations, $F=54.624$ $p=0.001$).”

Of the three pathways tested (PMK-1 p38/MAPK, TGF- β , and Insulin signaling), which were active under the conditions tested? If pathway is not active, then it is likely to expect that its knockout will have little effect on microbial composition, but overexpression may have consequences (possibly, no physiological). A fluorescent reporter for downstream genes (or qPCR of such genes), or direct DAF-16::GFP fusion may allow to answer this question easily, providing more mechanistic connection between the presence / absence of the effect for a given pathway.

Prior work (Berg et al. 2019) indicated that innate immune pathways were differentially regulated in worms raised on a complex soil microbiome; we have added the following text to clarify this point (**lines 59-61**): “All three pathways of innate immunity (p38, TGF- β , DAF-2/IGF) have been shown to be differentially expressed in worms raised on a complex microbiota as compared with *E. coli* (24).”

We concur with the reviewer’s comment that confirming induction of DAF-16 would strengthen the mechanistic argument, and thank the reviewer for the suggestion. **We have performed experiments using a DAF-16::GFP fusion line, which are now presented as Figure S3.** The results of these experiments are indicated in the text (**lines 71-72**): “We confirmed that microbial colonization under these conditions was associated with differential activation of *daf-16* using a fluorescent reporter assay (**Fig. S3**).”

January 6, 2021

Prof. Nic M Vega
Emory University
Biology
1510 Clifton Road
Atlanta, GA 30322

Re: mSystems00608-20R1 (Host immunity alters successional ecology and stability of the microbiome in a *C. elegans* model)

Dear Prof. Nic M Vega:

Both Reviewers found the revised manuscript to be significantly improved but identified several remaining issues that need to be addressed. Please constructively address all remaining reviewer concerns in revision. In particular, as suggested by Reviewer 1, tone down discussion of ecological succession in the Discussion (perhaps replacing with "colonization") since your resolution is limited to single timepoints.

Below you will find the comments of the reviewers.

To submit your modified manuscript, log onto the eJP submission site at <https://msystems.msubmit.net/cgi-bin/main.plex>. If you cannot remember your password, click the "Can't remember your password?" link and follow the instructions on the screen. Go to Author Tasks and click the appropriate manuscript title to begin the resubmission process. The information that you entered when you first submitted the paper will be displayed. Please update the information as necessary. Provide (1) point-by-point responses to the issues raised by the reviewers as file type "Response to Reviewers," not in your cover letter, and (2) a PDF file that indicates the changes from the original submission (by highlighting or underlining the changes) as file type "Marked Up Manuscript - For Review Only."

Due to the SARS-CoV-2 pandemic, our typical 60 day deadline for revisions will not be applied. I hope that you will be able to submit a revised manuscript soon, but want to reassure you that the journal will be flexible in terms of timing, particularly if experimental revisions are needed. When you are ready to resubmit, please know that our staff and Editors are working remotely and handling submissions without delay. If you do not wish to modify the manuscript and prefer to submit it to another journal, please notify me of your decision immediately so that the manuscript may be formally withdrawn from consideration by mSystems.

Sincerely,

John Rawls

Editor, mSystems

Journals Department
Reviewer comments:

Reviewer #1 (Comments for the Author):

The authors have nicely addressed some of my previous comments, including the added experiment with *daf-2*;*daf-16* double mutants (supporting previous assertions), and the demonstration of how different microbiota members are distinguished on plates. However, some comments are still not fully addressed, primarily: 1) lack of sufficient statistical support - claiming differences between microbiota of a certain mutant and that of wildtype worms (as the authors do in Fig. 2) should be supported by statistics in each case; using ANOSIM to show that overall, microbiotas of different strains are different, is not sufficient. 2) There was not sufficient (hardly at all) toning down of the discussion of ecological succession and the associated interpretations, which I think are misleading. The results more aptly relate to colonization, and I think that focusing on that would be more accurate and valuable. This includes the title, where "colonization and stability" would be in my view more accurate than "ecological succession and stability".

Furthermore:

with all the discussion of microbiota diversity and composition, what was left out was microbiota size - this is particularly relevant for Fig. 3, in which a relationship is presented between microbiota size and diversity, but the reader cannot relate size to time and host genetics (does microbiota size increase with age? is it larger in *daf-16* mutants and smaller in *daf-2* mutants?). The authors have the data, but presenting it as part of Fig. 3 (or a figure preceding it) would be important for interpreting results.

Line 48, Fig. S3 doesn't seem to be the right figure. The text addresses effects of host mutations on the composition of the microbiome, but the Fig. S3 I see shows CFUs in *daf-16* mutants.

Table S1 is missing footnotes. There is a legend, but the relation of which to the table is not clear (refers to A, B, and C, which do not appear in the legend, mentions 'm', not shown in table, and does not explain what is SP1 etc.). I still vote for showing the curves with R and p included in the panel; it would be far clearer.

In Fig. 2 the meaning of the circle should be explained in the legend.

"Due to the magnitude and bi-directional nature of the divergences from wild-type, we chose to use DAF-2/IGF mutant hosts in these experiments." - please rephrase to clarify your meaning.

Reviewer #2 (Comments for the Author):

The authors sufficiently changed the manuscript and answered my open questions. I think the new manuscript is now stronger and clearer.

There are some differences between the marked up manuscript and the final manuscript regarding the numbering of the supplementary figures. This also concerns the direct responses to the reviewer. This should be double checked.

I have one remaining minor point.

I wonder how countable MYb56 was. Fig. S1 shows now a picture of a mixed community on a plate where MYb56 covered the entire center of the plate. Was this counted as one colony? If so, is it sensible to assume that this is true? Otherwise, how was it counted? This should be addressed in the figure legend or methods section.

Reviewer comments:

Reviewer #1 (Comments for the Author):

In addressing the causes for inter-individual variation in microbiota composition, the authors take a quantitative approach to evaluate the roles of host metabolic state and immune system status versus the role of inter-species interactions among gut colonizers in affecting the composition of the gut microbiota. The study utilizes *C. elegans* as a model host, employing several mutants, and follows colonization from an eight-species bacterial community, members of which can be distinguished based on colony appearance and their colonization evaluated using CFU counts. The authors focus on the roles of the insulin signaling pathway, with mutants for DAF-16 - a central immune regulator, and for DAF-2, the insulin receptor, its negative regulator. They conclude that establishment of the gut microbiota and its ecological succession deviate from the neutral model. They show that host immunity imposes a constraint on bacterial colonization and on inter-species interactions, but that given this constraint, inter-species interactions, through ecological succession, play an important role in determining gut microbiota composition.

It is an interesting paper taking advantage of the authors' skills in quantitative analyses to extract useful information from a set of relatively straightforward experiments, suggesting new directions in utilizing a popular model organism to address fundamental questions in microbiome research. This study can be quite useful, but some glossing over potential caveats, unclarity in describing pivotal hypotheses (leading to results in Fig. 5), and over-interpretation of correlations as ecological succession dynamics weakens its conclusions. The authors should justify their choice of bacterial community, demonstrating that colonies could be efficiently distinguished in a mixed culture; consider inclusion a *daf-2;daf-16* double mutant control to enable focusing on DAF-16-dependent contributions of insulin signaling, which has broad effects on metabolism; acknowledge drawbacks relevant to some of the other mutants (e.g. VHP-1); and clearly distinguish between correlations between microbiota size and diversity in one time point and ecological succession along several time points. This may require toning down of conclusions, but could overall strengthen the paper. Detailed comments are shown below:

1. Methods: The synthetic microbiota in use is made of eight strains chose solely based on their colony appearance to enable easy scoring. One would expect that their interactions represent the most generalized ones possible, i.e. competition for resources (with this in mind, the extent of positive interactions identified in Fig. 4 is surprising; what kind of interactions could they represent?). Altogether, it should be acknowledged that this community is probably not quite representative of the richer milieu of interactions between bacteria and their hosts in a natural context.

We thank the reviewer for raising this point, as we believe this is an important consideration with regard to interpretation of these results. The reviewer is correct that this community is minimal, containing only eight strains, and for that reason extrapolation to larger communities should be done with caution. Additionally, this minimal community was selected from strains representing the worm native microbiome in part based on practical considerations. However, these are strains isolated from a study of the wild native microbiome of *C. elegans*, and may (or may not) have an

evolutionary history that would modify their interactions with one another and/or with the nematode host.

Further, taxonomic (and therefore presumably functional) diversity was another primary consideration in selecting these strains from the original bank of isolates. A recent publication by the lab where these species were isolated has characterized the functional diversity in these bacteria, indicating that metabolic competencies are predicted to be closely tied to taxonomy in these strains, and that higher-order interactions such as cross-feeding are observed, although the latter was tested in only a small subset of strains for which metabolic models could be constructed (Zimmerman et al. 2020, ref 21). We are in the process of characterizing the interactions among these species in more detail, and our results thus far are broadly consistent with expectations from Zimmerman et al. Based on these results, as well as our own experience with these strains, we believe that this minimal community, while small, is functionally diverse, and contains a sufficiently interesting diversity of interactions.

The text has been modified to indicate “These bacterial strains represent a taxonomically and functionally diverse subset of isolates from a wild *C. elegans* microbiome” and to cite the indicated reference (**line 33**). In the Discussion (**lines 264-268**) we have added the text “We chose to use a minimal eight-species community for tractability, and to select our community to represent a taxonomically and functionally diverse subset of the native host microbiome to allow for a range of possible interactions between species (20, 21). However, the small size of this community means that extrapolation to larger microbiota should be done with caution.”

In Figure 4, it is important to keep in mind that we are working with count data and not relative abundance. Positive correlations in these data do not necessarily reflect real positive interactions among bacteria – non-interacting or neutrally interacting bacteria can easily show these patterns when the number of bacteria per host varies across individuals. This is why we are able to get strong (but spurious) positive correlations in the simulated data in Figure S9 (formerly S8), despite the fact that the DMN model does not contain true positive *or* negative interactions between species. We have edited the text to reinforce this point, stating (**line 130-132**): “Note that positive correlations in these data need not reflect true positive interactions between species; a positive relationship between counts of two bacteria can simply reflect common species being common together, when the number of total bacteria varies among samples as it does in these data.”

To provide sound basis for all analyses, the authors need to better demonstrate that the choice of bacteria was working. Fig. S1 demonstrates how different colonies of the different community members look when cultured each on its own. However, this is not quite convincing. How, for example, can one count colonies of Myb56? These bacteria seem to give rise to a diffused lawn. Furthermore, in a mixed culture, competition may affect the size of adjacent colonies, as well as their color, hindering proper identification (Myb45, Myb120, and Myb181 might be indistinguishable). At the very least the authors should demonstrate easy distinction on a mixed culture plate.

We have added additional images to Supplementary Figure 1, demonstrating that colony morphologies remain distinct and are easily identified on community plates.

2. Claims about differences between microbiota composition in PCoA graphs should be supported by statistics, typically PERMANOVA, or ANOSIM, and would be easier to discern by including ellipses for the center of mass, with rays to the different microbiotas, as commonly done in such figures. Any thoughts about the nature of these differences?

We have updated the plots in Figure 2 to include ellipses for center of mass, and the results of statistical testing are described briefly in the text (**lines 50-51**): “All worm mutants were colonized under the same conditions used for N2 (above), and preliminary testing indicated differences between host strains (ANOSIM based on Bray-Curtis differences, 9999 permutations, $R=0.27$, $p<0.0001$; perMANOVA, 999 permutations, $F=54.624$, $p=0.001$).”

We hold that a convincing argument against the use of starburst plots for microbiome ordination data was made in Knights et al. 2014 (Rethinking “Enterotypes”, *Cell Host Microbe* 16(4):433-437), who indicated that these plots can suggest patterns where none exist. Further, due to the very large number of points in this data set, we believe that the extra plot components added by the starburst would make these plots excessively crowded and more difficult to read.

3. Ecological succession in the worm gut and the rules governing it should be cautiously interpreted. Yes, the idea that diversity decreases with community size makes sense, and has support in the literature. True, the authors identify a supporting correlation in D4 microbiotas. However, this is one time point, and not ecological succession per se. Furthermore, when the authors look how this trend holds when considering real ecological succession, i.e. D2, 4 and 6, the hypothesized rule is not well supported (at D6). All this should be acknowledged (better than as it is now hand-waved away on line 91) and be made clear rather than trying to present a stronger than possible support for rules governing ecological succession.

We thank the reviewer for pointing out our lack of clarity on this point – it was not our intention to actively support any particular theory of ecological succession with these data, merely to point out that our data are not inconsistent with the theoretical rule described here. As the reviewer states, these experiments were not designed as a test of diversity-size relationships and provide at best indirect (and host genotype-specific) evidence for any such relationship. For these data, the more important point is the difference in successional pattern in the different host genotypes. **We have therefore removed the statement in question.**

4. The competing hypotheses leading to the drop-out experiments shown in Fig. 5, (around line 165) are not clear, and the interpretation of the results of these experiments is not clear. This is an important part of the paper and should be well explained.

We concur with the reviewer on this point. The competing hypotheses underlying these experiments are rather technical and can be difficult to explain in a way that is clear to a broad audience. We have re-written the section in question to make the competing hypotheses more clear, as follows (**lines 180-195**):

“We expect this perturbation to have specific effects if there are substantial priority effects (where the early composition of the community affects the species-specific probability of success of later colonists) mediated by inter-species interactions. We should see very little effect in a highly stringent host (*daf-2*) where environmental filtering dominates, as we expect bacterial species to colonize based on ability to survive this environment rather than on ability to interact with other colonizers. Conversely, this perturbation may have a large effect in an uncontrolled host (*daf-16*) where interactions between bacteria, rather than interactions with the host, are the (theorized) dominant force driving community assembly.

Other outcomes are possible. If priority effects are mediated through changes in the host state rather than through interactions between bacterial species, this perturbation should produce smaller changes in environments where host control is of less relative importance (N2, *daf-16*) than in environments where the host is the dominant factor (*daf-2*). Alternately, if these systems are not driven by priority effects, the small perturbation introduced in these experiments should not alter community composition.”

5. To present the biological data, which is later-on abstracted in PCoA graphs, the authors should include in Fig. 1B the bar graphs representing microbiotas of all mutants.

These data are now presented as **Supplementary Figure 3A**.

6. The authors use a 15 min incubation with bleach to remove external contamination from worms. The bleach concentration is relatively low but the incubation is long. This is performed at 4C to stop intestinal peristalsis, presumably preventing bleach from entering the gut and affecting colonizing bacteria, but can they show that this works? Otherwise, microbiota composition might be biased by differential resistance to the leaking bleach.

We have validated the bleach protocol using highly sensitive *E. coli* DH5 α as a colonist of the worm gut, demonstrating that the protocol used here substantially reduces external bacteria without affecting intestinal counts of this easily killed bacteria. **These data are now presented in supplementary Figure S10D**. More anecdotally, we have used this bleaching protocol to sanitize larvae from contaminated plates; early instar (L1-L2) larvae do not transfer the contaminant after bleaching, while adult worms do, suggesting that the protocol works best for cleaning worms too small to carry contaminating microbes effectively in the gut.

7. Mutants in use:

DAF-2 is not only a repressor of DAF-16, although this is one of its more important and characterized roles. It affects metabolism as well, and this should be considered. Including experiments in *daf-2;daf-16* double mutants could serve as an important control for DAF-16-specific contributions of DAF-2. It should further be acknowledged that DAF-16 is not simply an immune regulator, but a regulator of many other stress responses, including oxidative stress, and xenobiotics.

We thank the reviewer for raising this point, and we concur that the double mutant control provides important additional information about the specific contributions of DAF-16 to

microbial community assembly. We have added a *daf-2(e1370);daf-16(mu86)* double mutant to our intestinal community data set. As the double mutant is sterile, it was necessary to make use of a mutant carrying an extrachromosomal array for propagation (CGC CF1449). The array is extremely unstable; non-GFP non-Rol animals, which have lost the extrachromosomal array, were very abundant in our samples and were easily sorted for single worm digests. As the *daf-2* and double mutant were best grown at 16C, we grew all comparison strains (N2, *daf-16*) to adulthood at 16C as well to provide appropriate controls, which provided further information about the likely role of *daf-16* in regulation of host-microbiome interactions in this system. **This information has been added to Methods and Table 1, and these data have been added to Figure 2 and Figure S4.**

We have altered the text in the Discussion to indicate the broad role of DAF-16 (**line 288**): “This pathway has been implicated in a broad range of stress responses (45, 46) including oxidative stress and xenobiotics in addition to response to bacterial pathogens...”

With that in mind, it's also worth acknowledging that DBL-1 signaling is also important to development, not only immunity.

Most studies on the role of *vhp-1* were performed with RNAi. Null *vhp-1* mutants are inviable. The *vhp-1(sa366)* strain is viable due to a weaker phenotype, and is probably a hypomorph, so it's not clear to what extent it leads to p38 over-activation. This should be acknowledged.

While it's understood that *glp-4* mutants should serve as a control for *pmk-1* mutants that share the same *glp-4* background, it should be acknowledged that *glp-4* disruption further affects protein translation, which may have additional effects. Furthermore, the *vhp-1* mutants in use do not share this background, which further complicates matters.

We have added text to the Discussion to clarify the limitations of the mutants used here (**lines 279-286**): “Interpretation of these results is complicated by the pleotropic nature of worm genetics. For example, DBL-1 is important in development as well as immunity (45). Disruption of *glp-4* is known to have broad effects, including changes in protein translation (46); the *glp-4* mutant was used here as a control for AU37, and the *vhp-1(sa66)* mutant used here for up-regulation of p38 does not have the *glp-4* mutation. Further, this *vhp-1* allele is probably hypomorphic (null alleles are lethal (47, 48)) with respect to p38 regulation. While our results suggest a complex role for innate immunity in regulation of the worm gut microbiome, further investigation of specific pathways is needed to disentangle immunity from other functions.”

Minor points.

1. Line 47: Saying that grinding mutants were not examined before for effects on microbiome composition is inaccurate. See Berg et al. 2019, for microbiota composition in *tnt-3* mutants. The results here should be compared to the previous ones.

We thank the reviewer for drawing our attention to this point of confusion. Here we intended to indicate that these specific mutants, which are mechanically different from those used in Berg et

al, had not been explored. However, as the conclusions are unchanged, we have rephrased the section in question to indicate that our results are consistent with those observed by Berg et al. **(line 56)**: “Increased permissiveness of the defective grinder did not substantially affect community assembly, consistent with previous results (24).”

2. Line 36, should be Fig. 1A.

3. Where are Figs 2E and 2F (line 63)?

We have corrected the figure references.

4. To improve the accessibility for the reader it would be helpful to split the results part into paragraphs with meaningful headers.

We have inserted headers into the Results section to break up the text into sections: *Effects of host genetics on microbiome assembly*; *DAF-2 signaling alters host control of microbiome assembly*; *DAF-2 signaling alters sensitivity of the microbiome to changes in colonization*.

5. Generally, please italicize gene names and species names also in graphs (e.g. Tab. 1, Supp Fig. 3, 4, 5...), and label axes understandably (e.g. x-axis of Supp. Fig. 10).

We have edited the indicated objects and legends to ensure correctness.

6. lines 110-115: A Spearman correlation coefficient was calculated for each worm strain but also bacterial strain (Supp. Fig. 6) but it is not clear which variables are correlated with each other. Please elaborate.

Spearman correlation coefficients were calculated for all pairs of bacterial species within each of the three host strains examined (N2, *daf-2*, *daf-16*); no correlations were calculated across host strains. We have edited the indicated text for clarity (**now line 124**): “To test this hypothesis, we first calculated the Spearman correlations for pairs of bacterial species within each of the N2, *daf-16*, and *daf-2* hosts”.

7. line 281: Please explain what is AXN.

An explanation, and the original citation, have been provided.

8. Fig. 3: It would be helpful to provide the correlation coefficient in the plot or the Fig. legend.

The summaries for all regressions are given in **Table S1**, as we now indicate in the figure legend, and we have moved the plot describing the linear fit to N2 data into Figure 3 (now Figure 3A).. Adding the entire set of coefficients and labeling these coefficients appropriately would produce

a significant amount of clutter due to the relatively large number of individual regressions in this set of plots. As these fits are merely descriptive and used here only to make confirmatory comparisons across host genotypes, we do not feel that the gain in information is worth the loss of clarity in this case.

9. Fig. 2, Supp. Fig. 3B: The colors of worm strains *dbl-1* and *ctIs40* are too similar so that the lack of change in microbiome composition between *ctIs40* and N2 (as stated in lines 61/62) is not apparent in the graph.

We have modified the color scheme to increase contrast among these worm strains.

10. Supp. Fig. 1: Please confirm that pictures were taken with the same magnification, or add a scale bar.

The images were taken on a handheld camera without scale bar capacity. **We have added multi-species plates to Figure S1** to provide context for how these colonies appear in proximity.

11. Supp. Fig. 2: Please explain the ten-worm moving window.

We have added explanatory text to the figure legend: "Running average of relative abundance data, calculated using a ten-worm moving window (average over all consecutive sets of ten worms in the ordered data set in A)."

Propagule pressure is likely not familiar to many. Adding an explanation would be useful.

We have added the following explanation to the text for clarity (**line 175**): "reduced propagule pressure (number of individuals of this species introduced per event, or per unit time)"

Reviewer #2 (Comments for the Author):

Comments to the author:

In this manuscript the authors show that host immunity has a strong effect on microbiome community composition and structure (in particular the interactions between the microbiota). Even though this is not the first study that shows the influence of *C. elegans* genetics on the microbiome, it is the first that shows a direct effect on bacteria-bacteria interactions and the effect of a perturbation on community assembly.

Specific comments:

Line 33: the authors claim that the community was selected to "represent the overall diversity of bacterial taxa in these samples [the *C. elegans* microbiome]", however, some of the "core" *C. elegans* microbiota are missing (e.g. *Pseudomonas*) and the diversity of natural *C. elegans* is

much higher. I understand that the selection was also based on colony morphology, which makes sense. I would still suggest to re-phrase the sentence.

We thank the reviewer for raising this point; the reviews are in accord on this, that it is necessary for us to clarify the grounds on which these bacteria were selected and to emphasize the functional diversity that is conserved by our selection process. We have re-phrased the sentence in question to indicate: “These bacterial strains represent a taxonomically and functionally diverse subset of isolates from a wild *C. elegans* microbiome (20, 21) (**Methods**). Each possessed a unique colony morphology when co-cultured on agar plates, allowing CFU counts for each species to be taken from mixed communities (**Fig S1**).”

Additionally, in Fig. S1 the picture of MYb56 does not support the claim that counting of individual colonies was possible. It would be good to see one that contains separate colonies.

We have provided additional images in Figure S1 to clarify the morphology of these bacteria and indicate the conservation of these distinct morphologies on community plates.

Fig. 1c: in this PCA the axis are labeled as Dim 1 and Dim 2, later they are labeled as PC1 and PC2. I would suggest to use continuously the same names.

We have changed the axis labels on all figures to consistently read “PC1” and “PC2”.

Line 36-37: I wonder how the frequency distribution of total colonization looked like. This would help to interpret Fig. S2. Did all these worms stem from the same well of the experimental plate or were the sampled from replicate wells (this should be also addressed in the methods section)?

We have added a histogram of the total CFU/worm data to Figure S2 to aid in interpretation of these data. There is clearly heterogeneity in colonization intensity, but the heterogeneity is not unstructured; a more complete description of the between-individual variation is beyond the scope of the present manuscript.

In this data set, N2 worms were sampled from twelve independent experiments conducted over the course of roughly a year. Each individual experiment contains data for 12-36 individual N2 worms taken from a single well. Other host strains follow a similar pattern and are represented by 2+ independent experiments where 12-36 individual worms were digested per strain. This information has been added to the **Figure 1 legend** and to the Methods (**lines 378-380**) to clarify the structure of the data.

Line 40: "Community composition changes over the course of colonization" should be re-phrased as individual worms were sampled (as stated in the Fig. legend).

The reviewer is correct that these data do not represent an auto-correlated time series, and we

concur that this is an important point with regard to these data. We have corrected the text to read **(line 43)**: “Community composition differs in worms sampled at different time points in colonization (Fig. 1C)”

Fig. 2: some of the colors are hard to tell apart, especially the yellow and brown in 2c. The same is true for Fig. 3b, 5a, f, k, d, i, n, and Fig. S9.

We have modified the color schemes in these figures to increase contrast among these worm strains.

Fig. S3: Could you please state the N for each of the mutants?

Sample size information for each of the mutants has been added to the figure legend **(now Figure S4)**.

Line 51: "...similar to N2 communities of the same size". Does Fig. S3 only show the community composition of worms that shared the same bacterial load?

The original statement should refer to the original figure S4 (now S5), showing the diversity-size relationships. However, in the interests of clarity, we have rephrased the indicated statement to read **(lines 54-56)** “While communities in the severe grinder mutant *phm-2* were large compared to N2 (median CFU/worm 36,190 vs 13,000), composition was within the range observed for N2 (Fig. 2A)”

Line 63: should be Fig. 2c

Line 64: should be Fig. 2d

Line 93: should be Fig. 3e

We have corrected the figure references as indicated.

Line 93-94: but the Shannon index is not decreasing. Here, it would be nice to have a graph that shows the differences in Shannon index for the different mutants. Fig. 3e should be 3f. It seems rather that the bacterial load is lower.

We thank the reviewer for bringing this to our attention; our original statement was inaccurate. We have re-stated the description in the text as **(lines 105-108)**: “*daf-16* hosts displayed large populations which continued to increase in size and diversity over the observed period (Fig. 3E, Fig. S6D-F), while *daf-2* hosts showed convergence to smaller microbiomes consisting mainly of three dominant bacteria (Fig. 3F, Fig S5G-I).”

Fig. 3: In the legend, (E) appears two times. E and F show a correlation between bacterial load and Shannon index. I wonder if the beta-diversity also changes over time (as it was shown for N2). For D, E, F: it looks as if there are differences in bacterial load between the mutants (especially *daf-16*), but it is hard to grasp from the type of graph. Is this difference statistically significant?

We have corrected the figure 3 legend.

The CFU/worm data in Figure S5 (**now Figure S6**; same data as summarized in Figure 3) show the trend in population size more clearly, as well as displaying the composition of these communities directly; we now refer the reader directly to this figure in the Figure 3 legend.

Additionally, we have added a violin plot of total CFU/worm counts to Figure S6. **The legend describes the results of statistical testing to compare total counts between groups, with p-value bins indicated on the graph.**

Fig. 4: what is the difference between Fig. 4 a-c and Fig. S6 b, d, f? Also, I find it easier to grasp the differences in the histograms shown in Fig. S6 than those in Fig. 4.

Only the programs used are different (Fig S6b,d,f were generated in Excel). Both use the same bin size, but make slightly different bin allocations; Excel takes bin numbers as upper levels, and R seems to use lower-level bins. The data are the same.

Line 250 ff: Please clarify if the developmental timing of the different mutants was controlled or monitored. Did they all develop in the same speed? If not, this, and how this might affect the results should be discussed, especially as *daf-2* mutants usually develop at a slower rate than WT N2. This would then switch the "mid succession" window that was only identified for N2.

The mutants do not all develop at the same speed. Part of troubleshooting these experiments was working out the timing of synchronizations. We sought to ensure that synchronized worms from different strains would reach adulthood at the same time so that they could then be colonized in parallel. Sometimes this required staggering synchronizations; in other cases, it was possible to group strains that grew similarly into experimental clusters. In the case of *daf-2*, for example, it was sometimes simplest to run replicates of this strain along with replicates of other slow strains such as *dec-1* (not shown – part of a separate project).

We have edited the Methods to indicate that this was the case (**lines 334-336**): “As some of the mutants used here develop to adulthood at different rates, care was taken to arrange and/or stagger synchronizations such that all strains within a given experiment reached adulthood at roughly the same time.”

Line 297: "constant cell densities" sounds as if the density was kept constant during the experiment, which, I think was not the case. This should be clarified.

The reviewer is correct; the bacterial metacommunities are started at a relatively high titer ($\sim 10^8$ CFU/mL), which helps to minimize change due to growth by limiting the number of bacterial generations in the liquid media, and are replaced every 48 hours to minimize change in the community and provide a continual source of live, fresh bacteria for colonization. However, there is some inevitable shift over time. We have replaced the word "constant" with "uniform" to indicate the initial conditions used.

Line 301 ff: Did you also perform biological replicates (in the sense that multiple wells contained the same mutant and community) or did all worms that were analyzed per mutant stem from the same well?

All worm strains in the intestinal community data set are represented by multiple (2+, usually 3+) biological replicates, conducted on separate days, with 12-36 worms taken per well per experiment depending on the total number of worm strains being processed in that experiment. (We use a 96-well mechanical disruption procedure, making it convenient to work in multiples of 12.) This is now reflected in the Methods (**lines 378-380**) and can be seen in the raw count data provided in Data Set S1.

Line 285 ff: As far as I understand all worms (with the exception of AU37 and *glp-4*) were constantly kept at 25{degree sign}C. Can you please clarify how this effected the *daf-2* (e1370) mutant as the dauer phenotype is temperature sensitive? This mutant develops into dauers when kept at 25{degree sign}C. Does that mean that dauers were analyzed? This should affect the colonization and would make it hard to compare it to N2 adults.

The *daf-2* strain was cultivated, and synchronized worms were raised to adulthood, at 16°C to prevent constitutive dauer formation; this is now reflected in the Methods (**line 331**, "Starved L1 larvae were transferred to 10cm NGM plates containing lawns of *E. coli pos-1* RNAi and incubated at 25C for 3 days (most) or 16C (*daf-2*; CF1449; N2 and *daf-16* when specifically stated) to produce reproductively sterile adults").

Figures 2 and S4 now contain a comparison of N2, *daf-2*, *daf-16*, and *daf-2*; *daf-16* adults raised to adulthood at 16°C. Briefly, it is apparent that the *daf-2* mutant exerts its effects on host-microbe interactions substantially through dysregulation of *daf-16*, and the effects of *daf-16* itself are dependent on the temperature at which larvae were raised.

Line 338 ff: I understand a PCA was based on Bray-Curtis distances? PCA is generally based on the Euclidean distance, but PCoAs can be used with Bray-Curtis distances, as far as I understand. The latter would additionally allow for statistical testing of a clustering of the groups of interest via perMANOVA, which would strengthen the results section.

The PCA is based on the log-transformed and standardized count data as indicated in the Methods. We initially used PCA here because, first, we could; these are non-sparse count data, suitable for Euclidean distance calculations after appropriate standardization. Second, use of PCA allowed us to establish the contribution of individual bacterial species to the ordination results; however, these results did not prove to be sufficiently interesting to justify exploring in the manuscript. Further, our choice of day 4 for sampling in the large data set meant that we were, with full forethought, not maximizing distances between host strains in these data; we expect false negatives in perMANOVA or equivalent tests, with respect to biological differences.

Although these experiments were not ideally designed for statistical tests of differences between groups, it is not incorrect to use this test on these data. **We have therefore edited the text to indicate the results of ANOSIM and perMANOVA testing based on host strain**, following calculation of the Bray-Curtis distances (**lines 50-51**): “All worm mutants were colonized under the same conditions used for N2 (above), and preliminary testing indicated differences between host strains (ANOSIM based on Bray-Curtis differences, 9999 permutations, $R=0.27$, $p<0.0001$; perMANOVA, 999 permutations, $F=54.624$ $p=0.001$).”

Reviewer #3 (Comments for the Author):

In this manuscript by Taylor & Vega, the authors explore the relationships between host genotype and microbiome. This study was carried out under controlled settings, using the microbiome of 8 species, which representing the diversity of the full microbiome of *C. elegans*. This paper adds to the existing literature and will be of interest to multiple labs that study *C. elegans* ecology, host-microbe, or host-pathogen interactions.

The authors combined canonical colonization assays with ecological approaches. For example they used the Shannon diversity index and also analyzed relationship between bacteria within the host, and the relationship between bacteria and host. The paper is generally well-written and easy to follow. The Analysis portion of Methods section is described sufficiently well for this process to be recapitulated by other labs.

I only have one major question: Of the three pathways tested (PMK-1 p38/MAPK, TGF- β , and Insulin signaling), do the authors know that any of these were active under the conditions tested? If the pathway is not active, then it is likely to expect that its knockout will have little effect on microbial composition. However, overexpression could easily have unexpected (and potentially artefactual) consequences. Fluorescent reporters for downstream genes (or qPCR of such genes), or direct DAF-16::GFP fusions may allow this question to be answered easily and provide more mechanistic connections between the presence / absence of the effect for a given pathway.

Minor points:

Check the manuscript for the consistent and standard use of the nomenclature: genotypes in

Table 1 or Fig 2 are not italicized. In Materials and Methods, one strain is referred to as AU37, and one next to it as glp-4, instead of SS104. Similar mixing is in Fig 2. *C. elegans* needs to be italicized (e.g. Fig 2 legend).

We have corrected the use of italics in figures, tables, and legends, and we have standardized our presentation of the strains in the Methods.

For Fig. 2, can statistics be generated to compare communities in different mutants?

We have edited the text to indicate the results of ANOSIM and perMANOVA testing based on host strain, following calculation of the Bray-Curtis distances (**lines 50-51**): “All worm mutants were colonized under the same conditions used for N2 (above), and preliminary testing indicated differences between host strains (ANOSIM based on Bray-Curtis differences, 9999 permutations, $R=0.27$, $p<0.0001$; perMANOVA, 999 permutations, $F=54.624$ $p=0.001$).”

Of the three pathways tested (PMK-1 p38/MAPK, TGF- β , and Insulin signaling), which were active under the conditions tested? If pathway is not active, then it is likely to expect that its knockout will have little effect on microbial composition, but overexpression may have consequences (possibly, no physiological). A fluorescent reporter for downstream genes (or qPCR of such genes), or direct DAF-16::GFP fusion may allow to answer this question easily, providing more mechanistic connection between the presence / absence of the effect for a given pathway.

Prior work (Berg et al. 2019) indicated that innate immune pathways were differentially regulated in worms raised on a complex soil microbiome; we have added the following text to clarify this point (**lines 59-61**): “All three pathways of innate immunity (p38, TGF- β , DAF-2/IGF) have been shown to be differentially expressed in worms raised on a complex microbiota as compared with *E. coli* (24).”

We concur with the reviewer’s comment that confirming induction of DAF-16 would strengthen the mechanistic argument, and thank the reviewer for the suggestion. **We have performed experiments using a DAF-16::GFP fusion line, which are now presented as Figure S3.** The results of these experiments are indicated in the text (**lines 71-72**): “We confirmed that microbial colonization under these conditions was associated with differential activation of *daf-16* using a fluorescent reporter assay (**Fig. S3**).”

March 17, 2021

Prof. Nic M Vega
Emory University
Biology
1510 Clifton Road
Atlanta, GA 30322

Re: mSystems00608-20R2 (Host immunity alters community ecology and stability of the microbiome in a *C. elegans* model)

Dear Prof. Nic M Vega:

Your manuscript has been accepted, and I am forwarding it to the ASM Journals Department for publication. For your reference, ASM Journals' address is given below. Before it can be scheduled for publication, your manuscript will be checked by the mSystems senior production editor, Ellie Ghatineh, to make sure that all elements meet the technical requirements for publication. She will contact you if anything needs to be revised before copyediting and production can begin. Otherwise, you will be notified when your proofs are ready to be viewed.

- Minimum resolution of 1280 x 720
- .mov or .mp4. video format
- Provide video in the highest quality possible, but do not exceed 1080p
- Provide a still/profile picture that is 640 (w) x 720 (h) max

We recognize that the video files can become quite large, and so to avoid quality loss ASM suggests sending the video file via <https://www.wetransfer.com/>. When you have a final version of

the video and the still ready to share, please send it to Ellie Ghatineh at eghatineh@asmusa.org.

Sincerely,

John Rawls
Editor, mSystems

Journals Department
Fig S2: Accept
Fig S6: Accept
Fig S7: Accept
Fig S4: Accept
Table S1: Accept
Fig S8: Accept
Fig S3: Accept
Fig S5: Accept
Data Set S1: Accept
Fig S1: Accept